# A comparison of patterns of microbial C:N:P stoichiometry between topsoil and subsoil along an aridity gradient

Yuqing Liu[1,2,a], Wenhong Ma[3,a], Dan Kou[2,4], Xiaxia Niu[2,3], Tian Wang[2,4], Yongliang Chen[5], Dima Chen[2], Xiaoqin Zhu[2], Mengying Zhao[2,4], Baihui Hao[2,3], Jinbo Zhang[1], Yuanhe Yang[2], Huifeng Hu[2] [*]

[1] School of Geography Science, Nanjing Normal University, Nanjing 210023, China

[2] State Key Laboratory of Vegetation and Environmental Change, Institute of Botany, Chinese Academy of Sciences, Beijing 100093, China

[3] Department of Ecology, School of Ecology and Environment, Inner Mongolia University, Hohhot 010021, China

[4] University of Chinese Academy of Sciences, Beijing 100049, China

[5] China Agricultural University, Beijing, 100083, China.

[a]These authors contributed equally to this paper.

**Abstract**

Microbial stoichiometry and its potential driving factors play crucial roles in understanding the balance of chemical elements in ecological interactions and nutrient limitations along aridity gradient. However, little is known about the variation in these features along aridity gradient due to the lack of comprehensive field investigations. Additionally, previous studies focused on the topsoil (0-10 or 0-20 cm); however, the minimum sampling depth for topsoil could impact the results of the vertical distribution of microbial stoichiometry. In the present study, we measured the variation in microbial stoichiometry, examined the major influential factors (climatic, edaphic and biotic factors) along an aridity gradient and determined whether the sampling depth affected microbial C:N:P stoichiometry. From the topsoil to the subsoil, the microbial C:N, C:P and N:P ratios varied from 6.59 to 6.83, from 60.2 to 60.5 and from 9.29 to 8.91, respectively. Only the microbial C:N ratio significantly increased with soil depth. The microbial C:N ratio significantly increased with increasing aridity in both topsoil and subsoil, while the microbial N:P ratio decreased along the aridity gradient only for the topsoil. This result implied that drought stimulated microbes tend to be more N conservative, especially those in topsoil. Among all the factors, the soil organic carbon (SOC) content and the fungi to bacteria ratio exerted the largest influence on the microbial C:N, C:P and N:P ratios at both soil depths, implying that the substrate supply and microbial structure together controlled the microbial stoichiometry. The results also revealed that the aridity index (AI) and plant aboveground biomass (AGB) exerted

negative impacts on the microbial C:N ratio at both soil depths, and the effects of AI
decreased in the subsoil. The results of this study suggested that the flexibility of the
microbial N:P ratio should be considered when establishing the sampling depth for
microbial stoichiometry study.
**Keywords**: grassland ecosystem, C:N:P stoichiometry, soil microbial biomass, aridity
gradient
**1  Introduction**
Ecological stoichiometry is a powerful tool for understanding the balance of chemical
elements required by organisms and the functions of ecosystems (Elser et al., 2000a;
Sterner and Elser, 2002). C, N and P are regarded as critical elements in global
biogeochemical cycling, and C:N:P stoichiometry in soil microorganisms offers
essential insight into the nutrient limitations of the organisms and communities in an
ecosystem (Manzoni et al., 2010; McGroddy et al., 2004).
A few studies have addressed the pattern of microbial stoichiometry along latitudinal
(Cleveland and Liptzin, 2007; Li et al., 2015; Xu et al., 2013) or environmental
gradients (Li and Chen, 2004; Li et al., 2012; Li et al., 2015). For example, Cleveland
and Liptzin (2007) analyzed microbial stoichiometry at the global scale and showed an
increasing trend in the microbial N:P ratio with higher latitude. However, Li et al. (2015)
summarized the data and found that the microbial N:P ratio decreased with latitude.
Undoubtedly, there is uncertainty in the values of microbial C:N, C:P and N:P ratios in
global studies, and the variations in these patterns might have been partially caused by
the different methods that were used in the various studies (Xu et al., 2013; Chen et al.,
2016). Furthermore, less exploration of soil microbial stoichiometry along an aridity
gradient at the regional scale impedes our ability to disentangle the trend of the changes
in microbial stoichiometry amid climate changes. Climate change, such as global
warming, is increasing the degree of aridity in drylands owing to the decreased
precipitation and increased evaporation (Wang et al., 2014; Li et al., 2017). Given this
background, key ecosystem processes that are regulated by soil microbes, such as soil
respiration and nutrient mineralization, may be dramatically impacted by the increased
degree of aridity, especially in fragile areas of arid and semiarid ecosystems
(Delgadobaquerizo et al., 2013; Chen et al., 2014). Therefore, we conducted a field
investigation across a 2100-km climatic transect in the Inner Mongolian grasslands to
determine how the microbial C:N, C:P and N:P ratios were affected by changing
environmental conditions.
Previous studies have also shown that a variety of abiotic factors impact microbial
C:N:P stoichiometry (Cleveland and Liptzin, 2007; Manzoni et al., 2010; Hartman and
Richardson, 2013). For instance, control experiments have found that warming can
indirectly affect the turnover of microbial biomass N by stimulating soil respiration
(Veraart et al., 2011; Butterbach-Bahl et al., 2013). Climate could exert an influence on
microbial stoichiometry through changes to the microenvironment, such as soil
moisture and temperature, and it could also impact the availability of substrates in the
soil (Nielsen et al., 2009). Moreover, edaphic variables, such as SOC (Maria et al., 2014;
Chen et al., 2016) and soil texture (Li et al., 2015), could be associated with nutrient
mineralization and availability, thus influencing the C:N:P stoichiometry in microbial
biomass (Griffiths et al., 2012). A labeled incubation experiment showed that the
mineralization of organic P was mainly driven by the microbial C demands in P-poor
soils (Aponte et al., 2010; Heuck et al., 2015). In addition, microbial C:N, C:P and N:P
ratios were also affected by biotic factors such as plant productivity and the composition
of the microbial community (Fanin et al., 2013; Chen et al., 2016). Generally, fungi
exhibit a higher C:N ratio than bacteria (Strickland and Rousk, 2010); thus, a shift in
the fungi to bacteria ratio is expected to result in microbial stoichiometry changes (Li
et al., 2012; Heuck et al., 2015). However, those findings were based on literature
analyses or small-scale experiments, and the variations in microbial C:N, C:P and N:P
ratios at the regional scale have rarely been assessed systematically, and the drivers of
these variations need to be addressed more specifically with appropriate experimental
designs. Moreover, most research has focused on the top 10 cm of soil, which often has
high C availability and nutrient contents. It can be assumed that the effects of potential
driving factors exhibit minimal differentiation at deeper soil depths. However, soil at a
deeper depth might contain microbial communities that are specialized for their
environment, and their functions may differ from the functions of the communities in
the topsoil (Fritze et al., 2000; Blume et al., 2002). Certainly, the drivers that are
responsible for the variations in microbial C:N, C:P and N:P ratios in deeper soil remain
poorly understood. Such knowledge of the nature of soil microbial stoichiometry is
fundamental for understanding ecosystem function, especially at the 10-20 cm soil
depth, which remains highly uncertain in the published studies.
Substrates for microorganisms, such as available nutrients and water, decline
exponentially with depth, and the top 20 cm of soil accumulates the greatest amount of
microbial biomass, thereby attracting the attention of most researchers (Fierer et al.,
2003; Xu et al., 2013). Soil at a 0-20 cm depth was regarded as the topsoil in some
studies, while other researchers divided the soil from 0-20 cm into different soil depths
to explore the vertical differences between these depths (Aponte et al., 2010; Peng and
Wang, 2016). However, most studies used 0-10 cm as the topsoil to facilitate sampling
and comparative research (Li and Chen, 2004; Cleveland and Liptzin, 2007; Chen et
al., 2016). The depth of topsoil varies among studies, and sampling depth can therefore
have impacts on the study of the vertical patterns in soil microbial stoichiometry
(Tischer et al., 2014). Given that soil represents a highly heterogeneous environment,
especially in terms of site-specific soil development history, it is difficult to draw
general conclusions (Xu et al., 2013; Camenzind et al., 2018). In addition, if a large
difference existed between the soil at 0-10 cm and that at 10-20 cm, microbial
stoichiometry would be underestimated due to the ambiguous limitation of topsoil
(Tischer et al., 2014). To identify the soil depth that is appropriate for sampling and to
improve the understanding of topsoil research at a global scale, we designed a study
that divided the topsoil into 0-10 cm and 10-20 cm depths to compare the differences
in microbial stoichiometry at the regional scale.
In Inner Mongolia grasslands, the aridity exhibits a gradient that increases from
northeast to southwest (aridity index, calculated as precipitation/potential
evapotranspiration, ranges from 0.16 to 0.54), thus providing an ideal platform to better
estimate the patterns and drivers of microbial C:N:P stoichiometry along an aridity
gradient (Chen et al., 2014; Li et al., 2017). In this study, we aim to access the effect of
soil depth on soil microbial C:N:P stoichiometry along aridity gradient. We
hypothesized that the microbial C:N and C:P ratios decrease and the microbial N:P ratio
increases with temperature (Cleveland and Liptzin, 2007; Li et al., 2015; Xu et al.,
2013), and the microbial C:N and C:P ratios decrease and the microbial N:P ratio
increases with decreasing aridity index (Wang et al., 2014; Li et al., 2017). In addition,
the identification of soil depth for vertical study is differernt in some published
literature (Li and Chen, 2004; Aponte et al., 2010; Tischer et al., 2014; Peng and Wang,
2016). We predicted that variation of bacterial and fungal taxa between soil depths
might contribute to the shifts in C:N:P stoichiometry, especially in the N:P ratio
(Mouginot et al., 2014; Camenzind et al., 2018). Therefore, we focus on (i) the effects
of potential driving factors on microbial C:N, C:P and N:P ratios in topsoil and subsoil
(ii) the response of the microbial C:N, C:P and N:P ratios to soil depth.
**2   Materials and methods**
**2.1   Study area**
This study was performed across the Inner Mongolian temperate grassland, which is a
central part of the Eurasian steppe. The study area is located at 39.2-49.6°N latitude and
107.8-120.1°E longitude and covers an area of 440,000 km$^2$. From northeast to
southwest, the mean annual temperature increases from -1.7 to 7.7°C, and the mean
annual precipitation decreases from 402 mm to 154 mm, approximately 80% of which
falls in the growing season from May to August . Three grassland types, meadow steppe,
typical steppe and desert steppe, are distributed along the northeastern to southwestern
gradient and are dominated by *Stipa baicalensis* and *Leymus chinensis*, *S. grandis*, and
*S. klemenzii*, respectively. The soil types corresponding to the three grassland types are
categorized as chernozems, kastanozems, and calcisols, respectively, according to the
soil classification system of the Food and Agriculture Organization of the United
Nations.
**2.2   Sampling and data collection**
Along this transect, a total of 58 sites that were slightly disturbed by humans and
domestic animals were sampled, including 10 in the meadow steppe, 28 in the typical
steppe, and 20 in the desert steppe (Fig. S1). Five 1×1 m$^2$ subplots were established,
one at each corner and one in the center of 10×10 m$^2$ plot. The plant community in the
subplots was identified, and the above-ground biomass (AGB) was harvested. At each
site, three replicate soil samples at depths of 0-10 cm and 10-20 cm were collected from
three 1 × 1 m subplots arranged diagonally in a 10 × 10 m plot. The samples were mixed
to form one composite sample. After gentle homogenization and removal of roots, the
soil was sieved through a 2-mm mesh and stored to conduct the further experiments.
The total carbon concentrations were measured using an elemental analyzer (Vario EL
Ш, Elementar, Germany). The soil inorganic carbon content was determined with a
carbonate content analyzer (Eijelkamp 08.53, Netherlands). The SOC content was
calculated by subtracting the soil inorganic carbon from the total carbon. The soil
elemental contents were reported in mmol · kg$^{-1}$. Soil pH was measured in a suspension
with a soil:water ratio of 1:2.5. After the removal of organic matter and carbonates, the
soil texture was determined using a particle size analyzer (Malvern Masterizer 2000,
UK).
**2.3 Aridity index**
The aridity index was extracted from the Global Aridity Index (Global-Aridity) dataset,
which provides high-resolution (30 arc-seconds or ~ 1km at the equator) global raster
climate data for the period 1950-2000 (http://www.cgiarcsi.org) (Zomer, Trabucco,
Bossio, & Verchot, 2008). The specific calculation formula is as follows:
$$\text{Aridity Index (AI)} = \text{MAP} / \text{MAE}$$
$$\text{PET} = 0.0023 \cdot \text{RA} \cdot (\text{Tmean}+17.8) \cdot \text{TD}^{0.5}(\text{mm/month})$$
where MAP represents the mean annual precipitation, obtained from the WorldClim
Global Climate Data (Hijmans et al. 2005); MAE represents the mean annual potential
evapo-transpiration (PET); Tmean represents the monthly mean temperature, TD is
calculated as the difference between the monthly maximum and minimum temperatures;
and RA represents the extra-terrestrial radiation on above the atmosphere.
**2.4   Soil microbial analyses**
Microbial biomass carbon (MBC) and microbial biomass nitrogen (MBN) were
determined following the chloroform fumigation-$K_2SO_4$ extraction method, according
to Vance et al. (1987) and Wu et al. (1990). The soil was preincubated at 25°C for two
weeks at a field water capacity of 40%. Then, the soil was fumigated with chloroform
for 24 h in a vacuum. The fumigated and nonfumigated samples were extracted using
0.5 M $K_2SO_4$ with a soil:solution mass ratio of 1:4. The C and N contents were
measured with a multi N/C analyzer (Anaytik Jena, Germany). Using a universal
conversion factor of 0.45 (Jenkinson et al., 1976), the amounts of MBC and MBN were
calculated by subtracting the amounts of extractable C and N in the nonfumigated
samples from those in the fumigated samples (Vance et al., 1987; Wu and Joergensen
et al., 1990; Joergensen et al., 1996). Microbial biomass phosphorus (MBP) was
estimated according to the method described in Hedley and Stewart (1982) and
modified by Wu et al. (1990). The fumigation procedure was the same as that for MBC
and MBN. The fumigated and nonfumigated samples were extracted using 0.5 mol·$L^{-1}$
$NaHCO_3$ and were analyzed to determine the total phosphorus concentration using a
colorimetric method. Using a universal conversion factor of 0.40, the amount of MBP
was calculated by subtracting the amount of extractable P in the nonfumigated samples
from that in the fumigated samples (Hedley and Stewart.,1982). Phospholipid fatty
acids (PLFAs) were extracted from the soil using the method described by Bossio and
Scow (1998). Briefly, 8 g of soil (dry weight) was used for PLFA analysis. The resultant
fatty acid methyl esters were separated, quantified, and identified using capillary gas
chromatography.The following PLFAs were used as markers for each of the specific
groups: for fungi, 18:1ω9c, 18:2ω6c, 18:3ω6c; for bacteria, i13:0, a13:0, i14:0, i15:0,
a15:0, 15:1ω6, 2OH16:0, i16:0, 16:1ω7c, 16:1ω9c, a17:0, i17:0, 17:1ω8c, cy17:0, i18:0,
18:1ω7, 18:1ω5 and cy19:0.
**2.5   Statistical analyses**
The C:N, C:P and N:P ratios in the soil microbial biomass were log10 transformed
before analysis to improve their normality (Fig. S3). Paired samples t-tests were used
to determine the differences in the soil microbial biomass C, N and P between the
topsoil and subsoil and the differences in the C:N:P stoichiometry ratios in the soil
microbial biomass. Ordinary least squares regression analyses were conducted to
evaluate the relationship between the C:N:P ratios in the soil microbial biomass and
latitude, aridity index, AGB, SOC, sand percentage and fungi to bacteria ratio (F:B
ratio). The analyses were performed with SPSS 19.0 software (IBM Corporation,
Armonk, NY, USA). A structural equation model (SEM) was used to test the
multivariate effects (direct and indirect) on the C:N:P ratios in the microbial biomass
through hypothetical factor pathways (Fig. S4). The SEM was constructed using the
Amos 17.0 software package (Smallwaters Corporation, Chicago, IL, USA).

## 3 Results

### 3.1 The variation in microbial C:N:P stoichiometry between soil depths along the environmental gradient

The results indicate well-constrained relationships among C, N and P in soil microbial biomass (Fig. S2). The soil microbial biomass C:N, C:P and N:P ratios varied by an order of magnitude. Significantly different water content, soil bulk density, sand percentages and SOC content were found between soil depths ($P < 0.05$, Fig. 1a, 1b, 1c, 1f). The microbial biomass C, N and P concentrations in the topsoil were significantly higher than that in the subsoil ($P < 0.05$, Table. 2). The C:N, C:P and N:P ratios in the microbial biomass of the topsoil were 6.59, 60.2, and 9.29, respectively, while those values in the subsoil were 6.83, 60.5 and 8.91, respectively (Table. 2). Moreover, the microbial C:N ratio in the subsoil was significantly higher than that those in the topsoil (Table. 2).

The results revealed a significant positive relationship between the AI and the microbial C:N ratio (Topsoil, $R^2 = 0.10$, $P < 0.05$; Subsoil, $R^2 = 0.09$, $P < 0.05$, Fig. 2a) and the decreasing trend between the AI and the microbial N:P ratio (Topsoil, $R^2 = 0.10$, $P < 0.01$; Fig. 2c). In addition, the decreasing trend was found between the microbial C:N ratio and MAT (Topsoil, $R^2 = 0.14$, $P < 0.01$; Subsoil, $R^2 = 0.10$, $P < 0.01$, Fig. 2d), while a significant negative relationship was found between the microbial N:P ratio and MAT (Topsoil, $R^2 = 0.19$, $P < 0.001$; Fig. 2f). The increasing trend between the microbial C:N

ratio and latitude was found in topsoil and significant positive relationships were found
in subsoil (Topsoil, $R^2$ =0.14, $P<$ 0.01; Subsoil, $R^2$ =0.12, $P<$ 0.05, Fig. 2g), while a
negative relationship was found between the microbial N:P ratio and latitude (Topsoil,
$R^2$ =0.18, $P<$ 0.001; Fig. 2i). The microbial C:N ratio was positively related to AGB
(Topsoil, $R^2$ =0.06, $P<$ 0.05, Fig. 3a), SOC (Topsoil, $R^2$ =0.12, $P<$ 0.01; Subsoil, $R^2$
=0.09, $P<$ 0.05, Fig. 3d) and was negatively related to the sand percentage (Topsoil, $R^2$
=0.11, $P<$ 0.01; Subsoil, $R^2$ =0.11, $P<$ 0.01, Fig. 3g). A significant positive relationship
was found between the microbial C:P ratio and the content of soil organic matter
(Subsoil, $R^2$ =0.08, $P<$ 0.05, Fig. 3e). No or only weak association was found between
the microbial C:N, C:P and N:P ratios and the AGB and F:B ratio in the subsoil (Fig.

257  3).

**3.2 Effects of potential driving factors on the microbial C:N:P stoichiometry at**
**topsoil and subsoil**
The final SEM adequately fit the data, as shown by several robust goodness-of-fit
measures (P value and minimum discrepancy). The model explained 38% (topsoil) and
27% (subsoil) of the variation in the microbial C:N ratio, 17% and 19% of that in the
microbial C:P ratio, and 29% and 16% of that in the microbial N:P ratio (Fig. 4a, b, c,
d, e, f). Effects of AI, AGB and SOC contenton the microbial C:N ratio were found at
both soil depths (Fig. 4a, 4b). The SOC content made the largest positive contribution
to the variation in the microbial C:N ratio in the topsoil (Fig. 4a, 4b). We found direct
effects of the sand percentage, SOC content and F:B ratio on the microbial C:P ratio at
both soil depths, and the SOC content made the largest contribution to the variation in
the microbial C:P ratio in the topsoil, which was higher than that in the subsoil (Fig. 4e,
4f). Influences of sand% and the SOC content on the microbial N:P ratio were found in
the topsoil, while the F:B ratio and the SOC content explained most of the variation in
the microbial N:P ratio in the subsoil (Fig. 4e, 4f).
**4   Discussion**
**4.1   The pattern of microbial C:N, C:P and N:P ratios along aridity and the**
**latitude gradient**
As stated in our hypothesis, the increase in the microbial C:N ratio and the decrease in
the microbial N:P ratio that were found along a temperature gradient in this study are
in accordance with the findings of Li et al. (2015) and Chen et al. (2016), who reported
similar variations in microbial stoichiometry along latitudinal gradient. Temperature
drives the variation in the growth of the microbial community, as high growth rates at
low latitudes require high RNA contents, causing the N:P ratio to decline (Chadwick et
al., 1999; Kooijman et al., 2009; Xu et al., 2013). In addition that, we observed that the
microbial C:N ratio significantly increased with increasing aridity index, while the
microbial N:P ratio decreased with increasing aridity index, indicating that drought
(decreasing aridity index) affects ecological stoichiometry by mediating the growth rate
of microorganisms in semiarid regions (Elser et al., 2000b; Peng and Wang, 2016). Dan
and Wang noted that increasing aridity reduced the soil microbial abundance in
drylands, and a decreased growth rate in dry areas might result in decreased allocation
to P-rich ribosomal RNA (and thus higher C:P and N:P ratios) (Wang et al., 2014;
Maestre et al., 2015). Additionally, microbial C:N ratio decreased with the decreasing
aridity index, which serves as a protective mechanism as microbes decrease their
nitrogen use efficiency (NUE, the ratio of N invested in growth over total N uptake)
and tend to be more N conservative under drier climatic conditions (Mooshammer et
al., 2014; Delgado-Baquerizo et al., 2017). Moreover, under drier climate conditions,
the soil microbial communities shift from acting as r-strategists (fast-growing
copiotrophs) to acting as K-strategists (slow-growing oligotrophs), as microorganisms
with K-strategies have lower nutrient demands (N and P) and growth rates, invest more
nutrients into extracellular enzymes to gain limited nutrients and thus have higher
cellular C:N:P ratios than r-strategists (Fierer et al., 2007; Fierer et al., 2010).
The microbial C:N ratio demonstrated an increasing trend with increasing latitude, in
contrast to the decreasing trend that was demonstrated for the microbial N:P ratio. Such
results paralleled the results of studies on ecological stoichiometry, which revealed that
the C:N ratio of microorganisms increased with latitude, while the N:P ratio decreased
with latitude, suggesting increasing N limitations in microorganism ecosystems in high-
latitude areas (Li et al., 2015; Chen et al., 2016). The regional scale microbial
stoichiometry followed the global-scale stoichiometry patterns that were observed for
plant leaves (Reich and Oleksyn, 2004; Yuan et al., 2011), litter (McGroddy et al.,
2004), and soil (Sardans et al., 2012), conforming to the substrate age hypothesis, which
predicts young soils to be N-limited, whereas old soils tend to be P limited (Walker and
Syers, 1976; Vitousek et al., 2010). Our study further illustrated the latitudinal pattern
of microbial stoichiometry and first attempted to examine the variation in microbial
stoichiometry along an aridity gradient at the regional scale.
**4.2    Direct effects of ecological factors on controlling microbial C:N, C:P and N:P**
**ratios at topsoil and subsoil**
Among the ecological factors examined, our study found that the patterns of microbial
C:N and C:P ratios were associated with SOC content and the F:B ratio, suggesting that
the available C and microbial community structure together regulated the shift in
microbial stoichiometry. If the environmental parameters were considered individually,
SOC content was found to be significantly positively related to the microbial C:N and
C:P ratios, which is consistent with the results observed from global data analysis,
suggesting that SOC content may control microbial stoichiometry by mediating the
substrate stoichiometry, e.g., the soil C:N and C:P ratios (Hartman and Richardson,
2013; Maria et al., 2014; Mooshammer et al., 2014). In deeper soil, microbial metabolic
processes are limited by C availability and energy (C), such as denitrification and P
mineralization, thus resulting in the effect of SOC on miccrobial C:N:P stoichiometry
in subsoil (Fierer et al., 2003; Peng and Wang, 2016; Camenzind et al., 2018).
SEM also illustrated that the microbial community structure is an important feature in
determining microbial stoichiometry. The F:B ratio has recently been found to have a
vital influence on the patterns of microbial C:N and N:P ratios in soil at a large scale
(Chen et al., 2016). An experiment indicated that fungi have lower resource
requirements and higher C:N and C:P ratios than bacteria; thus, the shift in the F:B ratio
impacted microbial C:N:P stoichiometry (Mouginot et al., 2014). In our study, the
lower F:B ratio might have led to a shift in the microbial nutrient stoichiometry at
deeper soil depths (Tischer et al., 2014). Overall, the SEM highlighted the important
role of the C availability and microbial community structure in driving the variations in
microbial C:N, C:P and N:P ratios at both soil depths.
Moreover, AGB and AI also exerted direct influences on microbial C:N or C:P ratios,
and those impacts mainly acted in the topsoil but were weaker in the subsoil. The
climate imposes important controls on both the plant community and the microbial taxa
along with their interactions with soil nutrients (Chadwick et al. 1999; Vitousek 2004;
Reich and Oleksyn, 2004). In particular, drier weather condition, and the decreasing
aridity index, could affect the growth and productivity of plants, and then shape a shift
in vegetation types along this grassland transect (Jaleel et al., 2009; Cherwin & Knapp,
2012). At the same time, the meadow steppe ecosystem with high productivity
maintained relatively high soil C and N contents, which resulted in high C:P and N:P
ratios in these regions; thus, plant productivity exerted a positive influence on microbial
C:N (Aponte et al., 2010; Manzoni et al., 2010). Because of the vertical distribution of
the influence of the AI, the effect decreased with soil depth.
Interestingly, our results revealed that the microbial N:P ratio was mainly impacted by

the F:B ratio and SOC content, while the sand percentage and SOC content had direct

negative effects on the ratio in the subsoil, suggesting the flexibility of microbial

stoichiometry in response to distinct resource supplies between topsoil and subsoil

(Peng and Wang, 2016). The soil depth affected the microbial biomass N and P, which

decreased nearly twofold from the topsoil to the subsoil (Table 2). However, the results

showed that the N and P cycles responded asymmetrically to soil depth, which might

be attributed to the high variability in P availability (Li et al., 2015; Zechmeister-

Boltenstern et al., 2016). Generally, P is mostly derived from parent material, while N

is mainly a biological element (Vitousek and Farrington, 1997; Vitousek et al., 2010).

Therefore, it is believed that P variations regulate large-scale patterns in microbial N:P

stoichiometry and nutrient-use strategies (Heuck et al., 2015; Camenzind et al., 2018).

With a high proportion of sand, the soil becomes porous, which may lead to increased

leaching of available P to deeper soil depths (Otten et al., 1999; Achbergerová and

Nahálka, 2011). Similarly, P leaching caused by weathering led to a shift in the N:P

ratio in the soil, and a vertical study found a high variation in the N:P ratio between soil

depths across a large scale (Tian et al., 2010). The high variability of the N:P ratio in

soil and soil microbial biomass therefore indicates that the N:P ratio could be an

indicator of the ecosystem nutrient status at deeper soil depths (Cleveland et al., 2007;

Chen et al. 2013; Li, et al. 2015).

**4.3 How deep should we dig to evaluate the topsoil the microbial stoichiometry**

**in vertical study?**

The results showed significant differences in the water content and SOC content
between topsoil and subsoil, suggesting that the resource supplies between topsoil and
subsoil were significantly different. We also observed that the microbial C:N, C:P and
N:P ratios varied between topsoil and subsoil, and significant difference was found in
microbial C:N ratio. Those results indicated that the flexibility of the microbial
stoichiometry responds to different resource supplies between soil depths (Tian et al.,
2010; Peng and Wang, 2016). Similar findings were found in the top 16 cm of soil in a
Mediterranean oak forest (0-8 cm and 8-16 cm), where the microbial nutrient ratios
(C:N, C:P and N:P) varied with soil depths (Aponte et al., 2010). Tischer et al. (2014)
sampled the top 20 cm of soil (0-5 cm, 5-10 cm, 10-20 cm) and observed that the
microbial C:N ratio changed with soil depth. Moreover, sampling to a depth of 10 cm
showed a significant difference in the microbial N:P ratio (Tischer et al., 2014). The
detection of the differences in the microbial N:P ratio in our study depended strongly
on the sampling depth, suggesting that the microbial N:P ratio might provide insight
into the nature of ecosystem nutrient limitations in a vertical study (Cleveland and
Liptzin, 2007; Fierer et al., 2010). In addition, SEM also showed that the microbial N:P
ratio was controlled by multiple potential driving factors at different soil depths,
indicating that a 0-10 cm or shallower sampling interval should be used when studying
the vertical patterns in the microbial N:P ratio.
**5   Uncertainties and perspectives**
The first uncertainty was related to the determination of fungal and bacterial biomasses
by PLFA markers, which have limited targets for fungi and bacteria. This uncertainty
should be noted when interpreting the results in the present study. Methodological
advances in sequencing approaches might be used to more accurately index the
microbial community and reveal insights into the regulation of microbial C:N:P
stoichiometry in distinct soil microbial taxa or functional groups. Second, the theory
used to construct the model is another source of uncertainty; the theory related to the
drivers of microbial stoichiometry used in this study was mostly derived from a
literature review and summarized data. In future research, more control experiments
with the manipulation of C availability, especially at deeper soil depths, would further
improve our understanding of the changes in microbial stoichiometry and nutrient
limitations under the impacts of global change.
**6    Conclusion**
The ratios of C, N, and P in the microbial biomass were 6.59:60.2:9.29 in the topsoil,
which deviated from the 6.83:60.5:8.91 ratio in the subsoil. Moreover, significant
differences were found in the microbial C:N and N:P ratios between topsoil and subsoil,
indicating that the flexibility of microbial stoichiometry should be considered for
vertical study. In addition, The trend of microbial C:N ratio increasing with aridity
index, consistent with the perspective that microbes mediate their nitrogen use
efficiency and tend to be more N conservative under drier climatic conditions. The
microbial N:P ratio trend along the aridity gradient was consistent with the growth rate
hypothesis that a decreased growth rate in dry areas results in decreased allocation to
P-rich ribosomal RNA and thus a higher N:P ratio. These findings confirmed the
importance of SOC content, the microbial structure and soil texture in shaping the
pattern of microbial stoichiometry in semiarid grassland systems. The influence of
ecological factors decreased from topsoil to subsoil, as well as the decline in climatic,
edaphic and biotic factors. Overall, these results illustrated N and P limitation in
microbial biomass at deeper soil depths along aridity gradient and limited responses to
ecological factors in the subsoil.
*Author contributions*. HH. WM and YY devised the study. YL carried out the
experiment and data analyses. DK, YC and DC assisted with the data analyses and
interpretation. XN, TW, XZ, MZ and HB assisted with the experiment. All authors
contributed to the preparation of the paper.
*Competing interests*. The authors declare that they have no conflict of interest.
*Acknowledgments*. We thank the sampling team members of Institute of Botany,
Chinese Academy of Science, for assistance with the field data collection.
*Financial support*. This work was supported by the National Basic Research Program
of China (2015CB954201 and 2014CB954303) and National Natural Science
Foundation of China (31470498)
**Table 1.** Basic information of study sites

| Biome | Latitude (° N) | Longitude (° E) | MAP (mm) | MAT (°C) | Aridity Index | AGB (g · cm$^{-3}$) | Dominant species |
|---|---|---|---|---|---|---|---|
| Meadow steppe | 48.1(43.9–49.6) | 119(116-120) | 353(262–381) | –0.45(–1.81–1.71) | 0.48 (0.38-0.54) | 136(88-168) | *Stipa baicalensis* *Leymus chinensis* |
| Typical steppe | 45.6(43.5-49.5) | 117(114-119) | 304 (205–402) | 1.11(-2.09-3.29) | 0.37 (0.25–0.50) | 102(49.4-159.8) | *Stipa grandis* *Stipa kryovii* |
| Desert steppe | 41.7(39.2-43.6) | 115(108-113) | 223(154–293) | 5.63(4.13-7.67) | 0.23 (0.16-0.32) | 43.4(24.6-76.5) | *Stipa klemenzii* *Stipa breviflora* |

Note: Data represent the means, with minimum and maximum values in parentheses. MAT, mean annual temperature; MAP, mean annual
precipitation; AGB, above ground biomass.

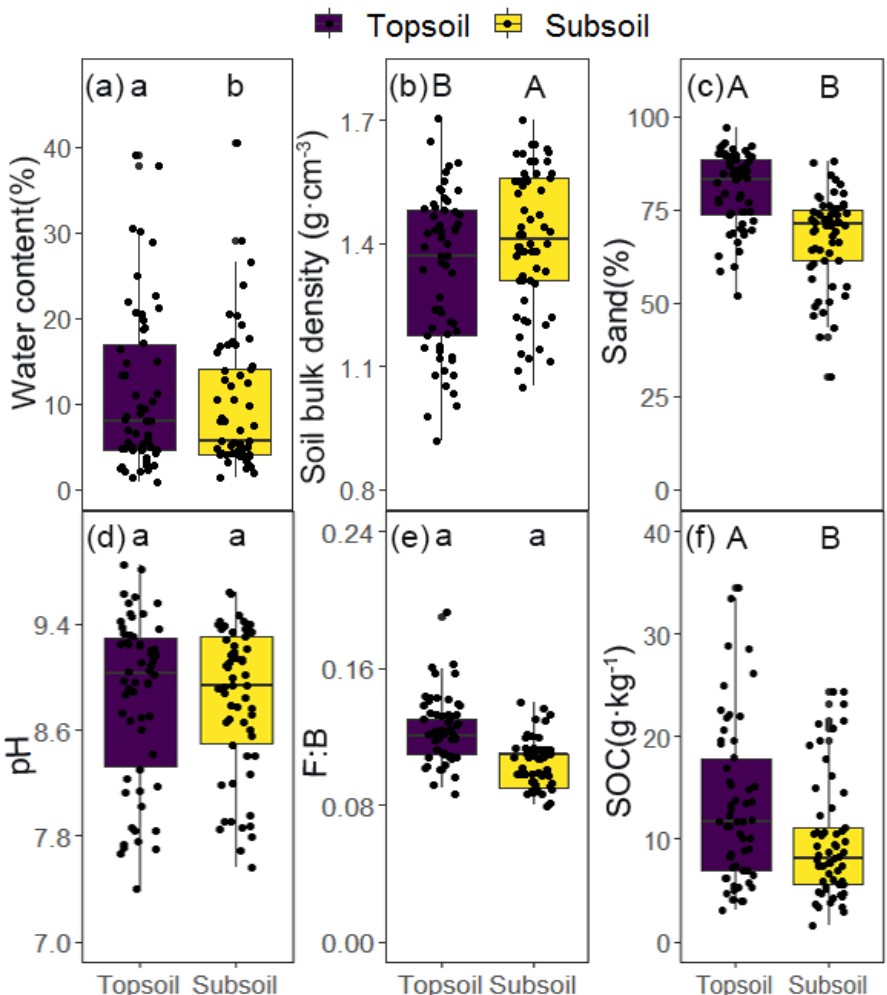


**Figure 1.** Basic characteristics of study sites across the Inner Mongolia grassland at different soil depths. Different letters indicate significant differences betwen soil depths on log10-transformed data (paired t-test, lowercase letter, $P<0.05$; uppercase letter, $P<0.001$)

**Table 2.** The microbial biomass C, N and P concentrations and microbial C:N:P stoichiometric ratios across the Inner Mongolian grassland at
different soil depths.

| Depth | MBC (mmol · kg$^{-1}$) | MBN (mmol · kg$^{-1}$) | MBP (mmol · kg$^{-1}$) | Microbial biomass | | |
|---|---|---|---|---|---|---|
| | | | | C:N | C:P | N:P |
| 0-10 cm | 21.8(18.5-25.1)A | 3.23(2.80-3.67)A | 0.38(0.32-0.44)A | 6.59(6.26-6.91)a | 60.2(55.6-64.8)a | 9.29(8.0-9.97)a |
| 10-20 cm | 14.5(12.4-16.6)B | 2.08(1.81-2.35)B | 0.24(0.21-0.27)B | 6.83(6.50-7.15)b | 60.5(56.0-65.1)a | 8.91(8.35-9.49)a |

Note:Different letters indicate significant differences between soil depths based on log10-transformed data (paired t-test, lowercase letter, $P<0.05$;
uppercase letter, $P<0.001$).

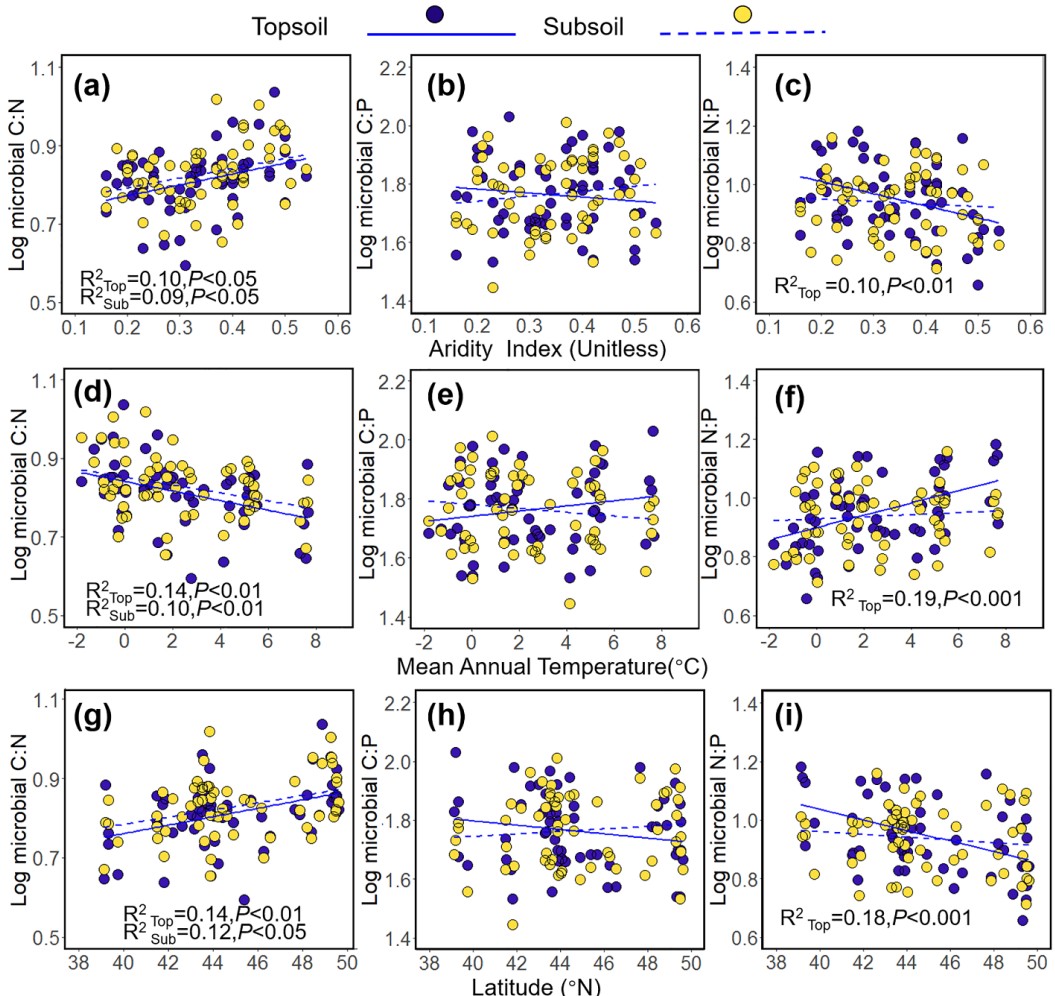


**Figure 2.** Relationships between the C:N, C:P and N:P ratios in soil microbial

biomass and aridity index (a-c), mean annual temperature (d-e) and latitude (g-i) in

the Inner Mongolian grassland.

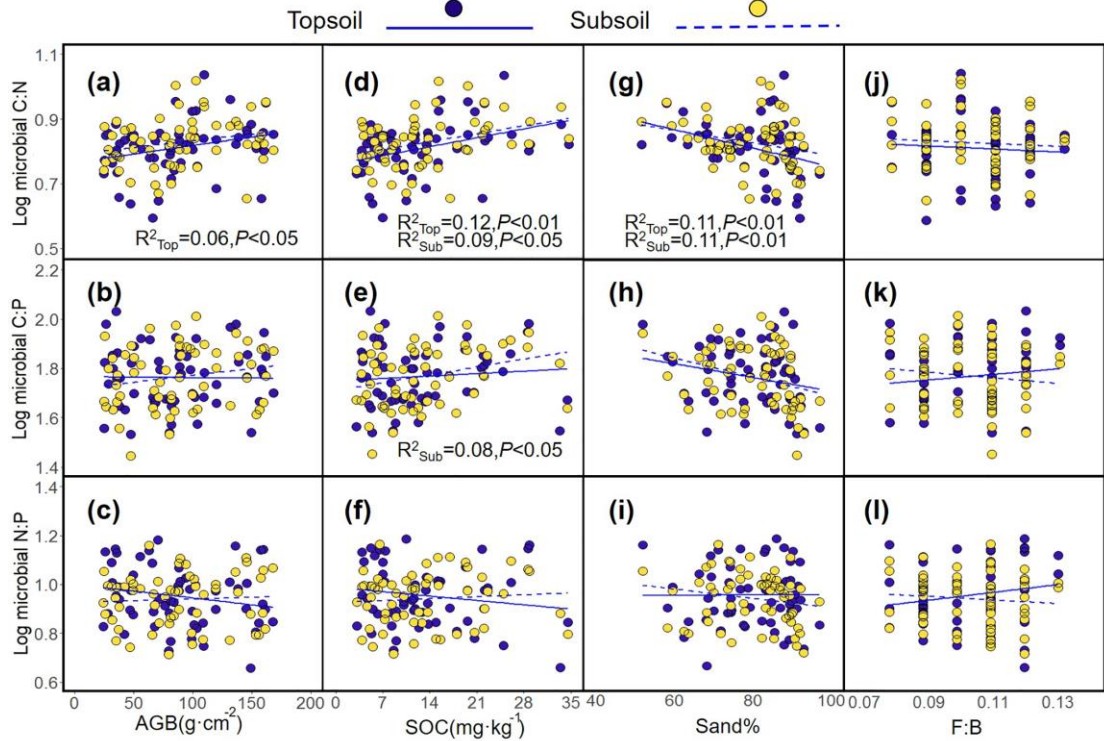


Figure 3. Relationships between the C:N, C:P and N:P ratios in the soil microbial

biomass and AGB (a-c), SOC (d-f), sand percentage (g-i) and F:B ratio (j-l). AGB,

above ground biomass; SOC, soil organic carbon; F:B ratio, fungi to bacteria ratio.

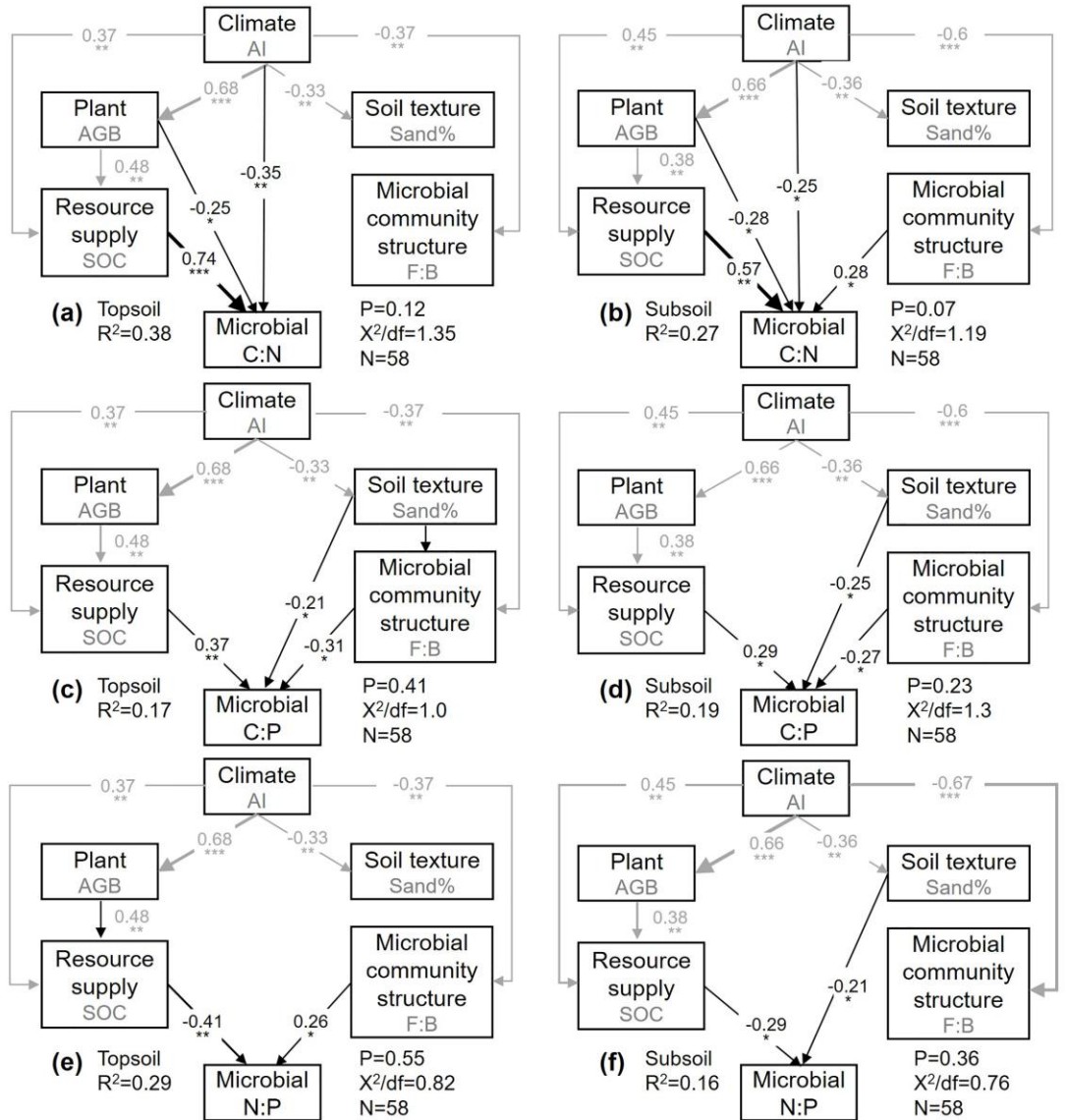


**Figure 4.** The structural equation model (SEM) shows the direct and indirect influences of various ecological factors on the microbial C:N (a,b), C:P (c,d) and N:P (e-f) ratios in the topsoil and subsoil. Black and gray arrows indicate direct and indirect pathways, respectively. Numbers on the arrows indicate standardized path coefficients, proportional to the arrow width. $R^2$ indicates the variation in the microbial C:N and C:P ratios explained by the model. *, $P<0.05$; **, $P<0.01$; ***, $P<0.001$.

 **Supplementary material**

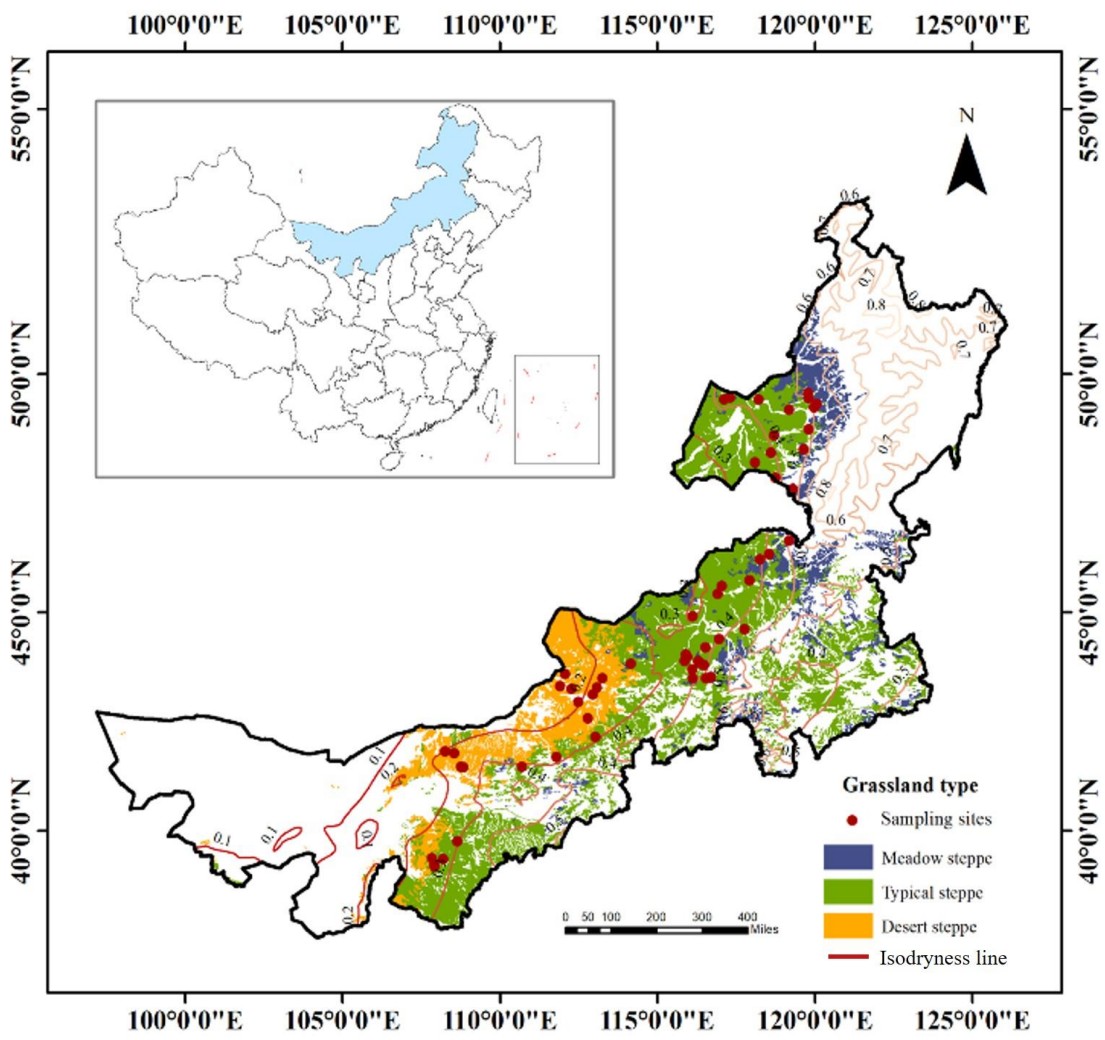


**Figure S1.** Geographic locations of the sampling sites in the Inner Mongolian grassland

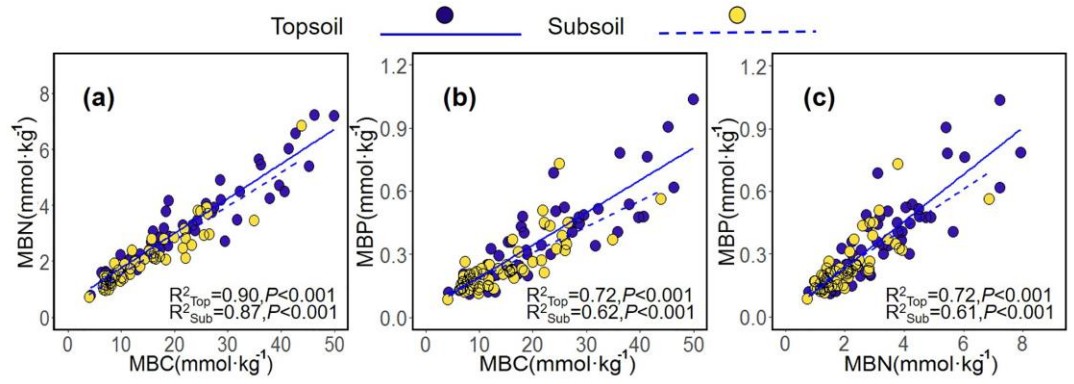


**Figure S2.** Relationships between the soil microbial biomass C, N and P concentrations

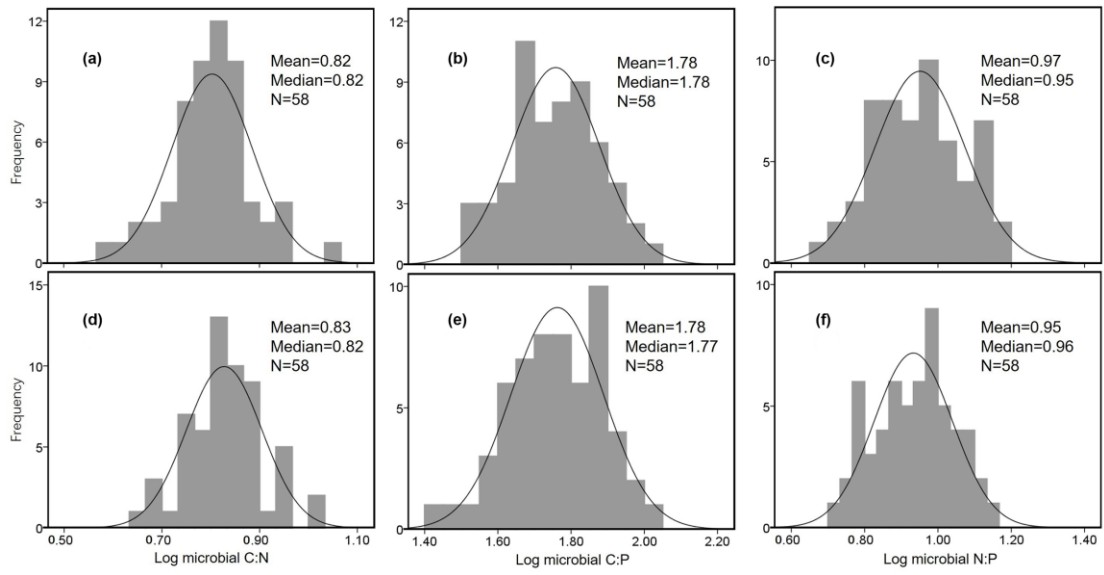


**Figure S3.** Histograms showing the frequency distributions of the soil microbial C:N,
C:P and N:P ratios in the topsoil (a-c) and the subsoil (d-f)

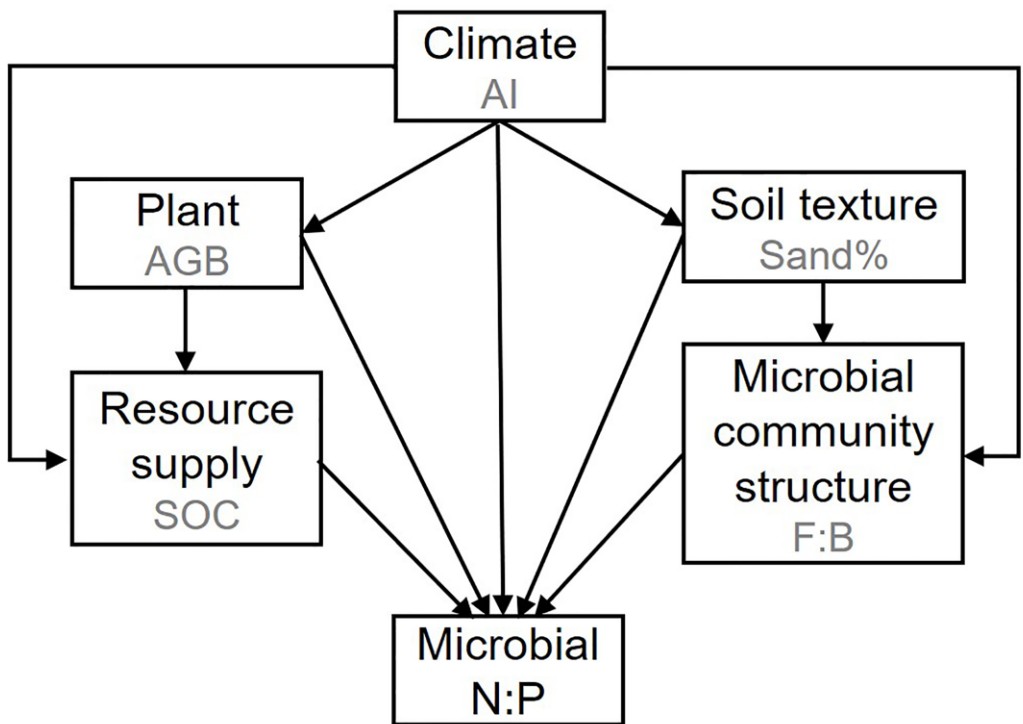


**Figure S4.** Hypothetical model showing how ecological factors affect microbial C:N:P
stoichiometry
**Table S1.** References to support the hypothetical models

| Pathway | Interpretation | Reference |
|---|---|---|
| SOC → Microbial C:N:P | Influence of SOC on microbial C:N:P stoichiometry | (Hartman et al., 2013; Maria et al., 2014; Mooshammer et al.,2014) |
| AGB → Microbial C:N:P | Plant necromass represents the fundamental resource for microbes to maintain element balance | (Cleveland et al., 2007; Aponte et al., 2010; Manzoni et al., 2010; Li et al., 2012; Zechmeister-Boltenstern et al., 2016) |
| AI→Microbial C:N:P | Influence of increasing temperature on microbial C and N cycle | (Wang et al., 2014; Zechmeister-Boltenstern et al., 2016; Chen et al., 2016) |
| Sand percentage → Microbial C:N:P | Influence of soil texture associated water-holding capacity and nutrient availability on microbial C:N:P ratios | (Cleveland et al., 2007; Xu et al., 2013; Maria et al., 2014; Li et al., 2015; Zechmeister-Boltenstern et al., 2016) |
| F:B ratio→ Microbial C:N:P | Influence of a shift in the composition of microbial community on microbial C:N:P ratios | (Ross et al.,1993; Cleveland et al.,2007; Aponte et al., 2010; Tischer et al., 2014; Zechmeister-Boltenstern et al., 2016; Chen et al., 2016) |

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

Strickland, M. S., and Rousk, J.: Considering fungal:bacterial dominance in soils –
Methods, controls, and ecosystem implications, Soil Biology and Biochemistry,
42, 1385-1395, https://doi.org/10.1016/j.soilbio.2010.05.007, 2010.
Tian, H. Q., Chen, G. S., Zhang, C., Melillo, J. M. and Hall, C. A. S.: Pattern and
variation of C:N:P ratios in China's soils: a synthesis of observational data,
Biogeochemistry, 98, 139-151, 2010.
Tischer, A., Potthast, K. and Hamer, U.: Land-use and soil depth affect resource and
microbial stoichiometry in a tropical mountain rainforest region of southern
Ecuador, Oecologia, 175, 375-393, 2014.
Trabucco, A., and Zomer, R.J. 2009. Global Aridity Index (Global-Aridity) and Global
Potential Evapo-Transpiration (Global-PET) Geospatial Database. CGIAR

624 Consortium for Spatial Information. Published online, available from the

625 CGIAR-CSI GeoPortal at: http://www.csi.cgiar.org.Vance, E. D., Brookes, P.

626 C. and Jenkinson, D. S.: An extraction method for measuring soil microbial

627 biomass C, Soil Biology and Biochemistry, 19, 703-707, 1987.s

628 Veraart, A. J., de Klein, J. J. M., and Scheffer, M.: Warming Can Boost Denitrification

629 Disproportionately Due to Altered Oxygen Dynamics, PLOS ONE, 6, e18508,

630 10.1371/journal.pone.0018508, 2011.

631 Vitousek, P. M. and Farrington, H. J. B.: Nutrient limitation and soil development:

632 Experimental test of a biogeochemical theory, 37, 63-75, 1997.

633 Vitousek, P. M., Porder, S., Houlton, B. Z. and Chadwick, O. A.: Terrestrial phosphorus

634 limitation: mechanisms, implications, and nitrogen-phosphorus interactions,

635 Ecological Applications, 20, 5-15, 2010.

636 Walker, T. W. and Syers, J. K.: The fate of phosphorus during pedogenesis, Geoderma,

637 15, 1-19, 1976.

638 Wang, C., Wang, X., Liu, D., Wu, H., Lü, X., Fang, Y., Cheng, W., Luo, W., Jiang, P.,

639 Shi, J., Yin, H., Zhou, J., Han, X. and Bai, E.: Aridity threshold in controlling

640 ecosystem nitrogen cycling in arid and semi-arid grasslands, Nature

641 Communications, 5, 4799, 2014.

642 Wu, J., Joergensen, R. G., Pommerening, B., Chaussod, R. and Brookes, P. C.:

643 Measurement of soil microbial biomass C by fumigation-extraction—an

644 automated procedure, Soil Biology & Biochemistry, 22, 1167-1169, 1990.

645 Xu, X., Thornton, P. E. and Post, W. M.: A global analysis of soil microbial biomass

646 carbon, nitrogen and phosphorus in terrestrial ecosystems, Global Ecology &

647 Biogeography, 22, 737–749, 2013.

Yuan, Z. Y., Chen, H. Y. H. and Reich, P. B.: Global-scale latitudinal patterns of plant

fine-root nitrogen and phosphorus, Nature Communications, 2, 344, 2011.

Zechmeister-Boltenstern, S., Keiblinger, K. M., Mooshammer, M., Peñuelas, J., Richter,

651          A., Sardans, J. and Wanek, W.: The application of ecological stoichiometry to

plant–microbial–soil organic matter transformations, Ecological Monographs,

85, 133-155, 2016.
