# Peer review of "A comparison of patterns of microbial C:N:P stoichiometry between"

_Biogeosciences, 2019_

## Referee Comment (RC1) · Anonymous Referee #1 · 25 Nov 2019

General comments: This manuscript (MS) examined soil microbial C:N:P stoichiometry along a large aridity gradient across different temperate grassland biomes, using 58 plots sampled from a 2100-km transect in Inner Mongolian, China. The dataset is good not only in that studies of soil microbial stoichiometry along great aridity gradient is still limited, and also because they examined difference in patterns and potential drivers between top and subsoil. The MS is generally well written, though there were some clear typewriting flaws and some sentences not easy to understand. I suggest a minor revision, mainly on the improving of the statistical analyses, and clarity and

readability of the MS.

Specific comments: The title: from the title I have first thought that you have sampled much deeper than 10 cm. However, I then realized that you have sampled to a depth of 20 cm. I would suggest to revise the title and delete "How deep do we dig for surface soil?". The rest part of the title is good enough.

Methods: In 2.2, how the above-ground biomass data was obtained was not clarified, but this data was used in statistical analyses. In addition, the method to calculate aridity index need to be introduced. Though the data was extracted from database, the Equation needs to be introduced for readers to better understand the biological meanings. Further, there were several different indices for aridity.

Statistical analyses: The plots were sampled from a northeast to southwest transect, which include variations in both temperature and aridity. It remains unclear how temperature contribute to the geographic patterns reported here. Considering the large difference in aridity from typical steppe to desert steppe, personally I agree with your results on the role of aridity on microbial stoichiometry. However, you may consider to include temperature as a predictor, at least in bivariate analyses, to make your conclusions more robust.

Results and Discussion: The SEM showed that climate have indirect effects through AGB, SOC and F:B. These are also interesting results, but was not mentioned in results. I also suggest to added some discussions of these indirect effects, though some of them were mentioned in discussion implicitly. Anyway, these indirect effects are part of the full picture how climate affects soil microbial stoichiometry.

By the way, the MS used many abbreviations, which markedly decrease the readability. Please try to remove unnecessary ones (e.g. MS, TS, DS in Table 1)

Minor comments: L45: meaning not very clear.

L58: A few studies. You have listed some studies along latitudinal and environmental

gradients in the subsequent text.

L62, 63: with higher latitude?

L64: replace values with patterns

L142: and at a depth of 10 cm, what does this mean?

L 171: aridity index (AI)

L217: "were well constrained (Fig. A2)". It needs to be explained.

L219: larger->higher

L223: the microbial C:N ratio in the subsoil was significantly higher than that in the topsoil (Fig. 2b). This result can not be found in Fig 2b. I guess it was in Table2.

L238 (and elsewhere) Effects of potential driving factors

L303: microbial C:N:P stoichiometry impacted the microbial community structure as a result of the F:B ratio. Do you mean that C:N:P stoichiometry affects microbial community structure? This seems you be conflict with the SEM. In SEM (and the sentence in line 300ïij■303), the logic is that C:N:P stoichiometry changes as a result of community structure.

Fig. 1 Where are the difference among the biomes here? In the caption (also in that of Table 1), you mentioned: MS, meadow steppe; TS, typical steppe; DS, desert steppe. But you did not show the results at all.

Table 2: across 404 the Inner Mongolian grassland at ??? Biome: soil depth? You did not compared biomes here in the Table.

Fig. 3: Why the figures of F:B were different from others? It seems many plots have a same F:B value.

---

## Referee Comment (RC2) · Anonymous Referee #2 · 25 Nov 2019

General comments

This paper focus on the sampling depth for analysing microbial stoichiometry C, N and P at 0-10 cm or 10-20 cm soil depth. It is an interesting study made in permanent grassland with an aridity index, but the interpretation and presentation of the data should be improved. The paper will profit from clearer hypothesis that can be tested, and more clear wording and presenting of the results. I do also suggest putting the correlation analyses given in 3.1 into a table, which would make it more accessible for the reader. Your question: "How deep do we dig for surface soil?" Should be clearly answered in

the conclusion.

Specific comments

Normally subsoil is used for the soil under the surface soil/ topsoil that are less affected by plant roots and tillage operations. However, I assume there were no tillage at the sites referred to in the present paper. The root distribution and rooting depth for the different sites are not given, but in permanent grassland most of the rooting and microbial activity is in the upper soil layer. I would still be reluctant to use the "subsoil" as a term for the soil layer at 10-20 cm depth as the roots would likely go deeper than 10 cm. Surface soil and topsoil are in many cases used as synonyms and the heading is therefore confusing. I suggest in stead: How deep do we dig for surface soil? A comparison of patterns of microbial C:N:P stoichiometry between an upper and lower soil layer along an aridity gradient.

Hypotheses

When you present hypotheses, it should be possible to test them and to either confirm or reject them and the result of the testing of the hypotheses should be clearly presented in the conclusion.

(i) microbial C:N and C:P ratios increase and the microbial N:P ratio decreases across an aridity gradient because of differences in nutrient-use efficiency. The first part of this hypotheses "microbial C:N and C:P ratios increase and the microbial N:P ratio decreases across an aridity gradient", you have actually tested in the present paper, but the result is not clearly written in the conclusions. In Figure 2, C:N, C:P and N:P ratios are given along an aridity Index (Gradient). Because of the very low relationships between the ratios and the aridity index, this part of the hyphothesis cannot be confirmed. $R^2 = 0.1$ is very low. In discussion you write: "microbial C:N and C:P ratios increase and the microbial N:P ratio decreases across an aridity gradient". I do not agree with this statement. Because of the low $R^2$, a $P < 0.05$ does not say much. If you look at figure 2, you see that the variation in within C:N and C:P sites at the same aridity is much

larger than the impact of Aridity index. I would rather call it a trend, then to state a significantly impact. The second part of the first hypothesis "because of differences in nutrient-use efficiency", you do only discuss and do not test. I would leave that out from the hypothesis.

(ii) Due to variations in resource supply among different soil depths, the effects of driving factors on microbial C:N, C:P and N:P ratios might decrease with soil depth. This hypothesis you have not tested and cannot do, as you do not know if variations in resource supply among different soil depths actually do effect driving factors on microbial C:N, C:P and N:P ratios. What you can test is: "Microbial C:N, C:P and N:P ratios do vary with soil depth." In the results 3.1 lines 222 to 223 you write: "Moreover, the microbial C:N ratio in the subsoil was significantly higher than that in the topsoil (Fig. 2b)." I assume you must mean table 2? If this is the case, such a hypothesis could be confirmed for C:N ratio, and rejected for C:P and N:P ratios. Obs,. You write in the abstract (line 32-34) :" We found that the microbial C:N , C:P and N:P ratios varied with soil depth.Âż According to table 2, they do not.

(iii) to adapt to the imbalance of resources, microbial C:N, C:P and N:P ratios vary between soil depths and at a depth of 10 cm, which could influence the research on the vertical patterns of microbial stoichiometry. I do not understand what you mean by this hypothesis. You should convert it to a hypothesis that can be tested and clearly present the result of the hypothesis. Do you mean "Microbial C:N, C:P and N:P ratios do vary with soil depth. At 0-10 cm depth the ratios are more influenced of an aridity gradient and other ecological factors than at 10-20 cm soil depths"?

In 3.1 you refer to "environmental gradient" in the title, but you do not refer to what you mean with "environmental gradient". You do focus on the impact of Latitude, but I do not understand for which purpose. And again the degree of explanation is low (R2= 0.14) and the variation is large.

Because this study is done on three grassland types (meadow steppe, typical steppe

and desert steppe) with corresponding soil types, I do miss the discussion on impact of grassland types, plant roots and rooting pattern on the microbial stoichiometry. Because aridity gradient (index) is central in this study it should be given how it was calculated (Line 171-172).

Figure A3 need some introduction. How did you develop this?

Technical corrections Line 181 and line 189, You must explain what a universal conversion factor is, what the units are and give a reference to where you got it from. Line 185 Which principal method is used? Cloroform fumigation? Hedley and Stewart (1982) is not given in the reference list. Line 201 Was the log10 transformed ratios normally distributed? Line 223 , Should it be table 2, not Fig. 2b?

Please also note the supplement to this comment:
https://www.biogeosciences-discuss.net/bg-2019-351/bg-2019-351-RC2-supplement.pdf

---

## Referee Comment (RC3) · Anonymous Referee #3 · 7 Dec 2019

The manuscript requires further clarification on methods, resolution of data and a more realistic presentation of the data analysis. The study design cannot answer the title of the paper, the 'gradient' variables are poorly described, and the methods are lacking with respect to the most important 'variables'. To some extent, the required revisions are minor. However, the focus on 'depth' in the manuscript title suggests that the authors need a major revision with respect to their study hypothesis. Please see the specific comments for further direction on the required revisions.

Title: you cannot answer the question 'How deep do we dig for surface soil'? Because

you did not dig very deep / or a high dig with high incremental accuracy. Two 10 cm samples do not answer the question. L42 is influenced the correct term? What was the relationship? L46-48 delete last sentence. Unsupported. L87 why 'might be'? L109 revise wording 'climate change background'. This study does not truly address deeper soils. L132 it this truly an 'ideal' platform. The resolution of the resolution of the aridity index is less than ideal. L135 at two depth. . . why especially in the surface. This is common? L150 what is the proportion of snow? L158 define slightly? Agricultural? Heavy grazing? Infrequent grazing? L159 why the uneven sample numbers per grassland type? Was this weighted by area? L161 what stop at 20 cm? L161 where were the three plots sampled? Corners and centre? L164 sentence is incomplete. L168 what elemental contents? Carbon only? What are the other elements? If other elements, how were they measured? L170 how was organic matter and carbonates removed from the soil? Carbonates should not be removed before texture is estimated. They are part of the mineral soil texture. L172 what was the resolution of the AI database? Is this adequate to evaluate against site specific measurements? If the metric is important why not calculate at each site? L172 what about bulk density? How was it measured? Reported? Why not use loss-on-ignition? How was AGB biomass measured. This is not explained but is an important measure (as indicated by the abstract) L180 is that ration based on mass or volume? L193 what different phases? L201 why t-test? Maybe an ANOVA should be used to account for the different grassland types? Or was a t-test applied to each type? If the latter, was the p value corrected for multiple tests? L206 AGB is not defined. L206 provide more details on the source of AL and AGB. What is there resolution? Is there a gradient in the data? Demonstrate that they are gradients. How are they estimated / measured? Provide a description of the data in the results (if they are important variables). L214 what is the gradient? L216 does distinct mean 'significantly different' L216 why is bulk density mentioned here. . . and only here? How was it measured? Did it differ greatly between grassland types? Why was soil microbial biomass not weighted by bulk density? L218 the concentrations were larger but was the pool larger? Use the bulk density to evaluate the pool difference. L225-235

these are very weak significant relationships. This should be acknowledged. Similarly, the relationships in Figure 2 and Figure 3 are not very convincing of a relationship (s). L225-235 you are regression carbon against a ratio that contains carbon... this is spurious? L237 clarify... subsoil is reported in L232, L233, and L235. L274 drought? Clarify. L285 many things change across latitude. Is microbial biomass influenced by latitude or the change in grassland type / climate / etc. Will microbial biomass also change across longitude? What is the range in the aridity gradient in the current study? L292 this is essentially stating that carbon is related to a ratio that includes carbon. This is not surprising. Is this a spurious (correlation) regression? L309 how were AGB and AI measured? Are they site specific or regional indicators? They only show a weak relationship with little predicative power. L334 did you quantify spatial heterogeneity? How? L337 you cannot answer this question. L341 are the pools distinct? L350 you tested limited depth, with course increments. L369 what about pools? L375 not shown, this statement is too strongly with respect to drought. There was a weak relationship using a coarse metric. L383 edaphic? Influence of soil on soil? L384 you need to demonstrate the gradient

---

## Author Comment (AC1) · 8 Jan 2020

Anonymous Referee #1 General comments: This manuscript (MS) examined soil microbial C:N:P stoichiometry along a large aridity gradient across different temperate grassland biomes, using 58 plots sampled from a 2100-km transect in Inner Mongolian, China. The dataset is good not only in that studies of soil microbial stoichiometry along great aridity gradient is still limited, and also because they examined difference in patterns and potential drivers between top and subsoil. The MS is generally well written, though there were some clear typewriting flaws and some sentences not easy to understand. I suggest a minor revision, mainly on the improving of the statistical analyses, and clarity and readability of the MS. Response: Thanks for your positive comments. We have carefully revised our manuscript according to your suggestions. Please see more details in our reply to your specific comments.

Specific comments: The title: from the title I have first thought that you have sampled much deeper than 10 cm. However, I then realized that you have sampled to a depth of 20 cm. I would suggest to revise the title and delete "How deep do we dig for surface soil?". The rest part of the title is good enough. ResponseïijŽThanks for your constructive comments. Following your suggestion, we have revised title as "A comparison of patterns of microbial C : N : P stoichiometry between topsoil and subsoil along an aridity gradient".

Methods: In 2.2, how the above-ground biomass data was obtained was not clarified, but this data was used in statistical analyses. In addition, the method to calculate aridity index need to be introduced. Though the data was extracted from database, the Equation needs to be introduced for readers to better understand the biological meanings. Further, there were several different indices for aridity. Response: Thanks for your suggestions. We are sorry that we missed this information. We have included more details on the methods of data extracting in the revised manuscript. We measured the aboveground biomass by harvesting the aboveground part of the plants in the sampling plot. We have revised as" Thanks for your suggestions. We are sorry that we missed this information. We have included more details on the data extraction and data acquisition methods in the revised manuscript. We have revised it as "Aridity index was extracted them from the Global Aridity Index (Global-Aridity) datasetïijŇwhich provide high-resolution (30 arc-seconds or $\sim$ 1km at equator) global raster climate data for the 1950-2000 period (http://www.cgiarcsi.org) (Zomer, Trabucco, Bossio, & Verchot, 2008). The specific calculation formula is as follows: Aridity Index (AI) = MAP / MAE PET=0.0023ÂůRAÂů(Tmean+17.8)ÂůTD0.5(mm/month) where MAP represents

mean annual precipitation, obtained from the WorldClim Global Climate Data (Hijmans et al. 2005); MAE represents mean annual potential evapo-transpiration (PET); Tmean represents monthly mean temperature, TD is calculated as the difference between monthly maximum and minimum temperatures; RA represents the extra-terrestrial radiation on top of atmosphere.

Statistical analyses: The plots were sampled from a northeast to southwest transect, which include variations in both temperature and aridity. It remains unclear how temperature contribute to the geographic patterns reported here. Considering the large difference in aridity from typical steppe to desert steppe, personally I agree with your results on the role of aridity on microbial stoichiometry. However, you may consider to include temperature as a predictor, to make your conclusions more robust. Response: thank you for your helpful suggestions. We have added related figure, result and discussion in the manuscript. Figure 2. Relationships between the C:N, C:P and N:P ratios in soil microbial biomass and latitude (a-c), mean annual temperature (d-e) and aridity index (g-i) in the Inner Mongolian grassland. In result: Besides, significant negative relationships were found between the microbial C:N ratio and MAT (Topsoil, $R^2$ =0.14, $P<0.01$; Subsoil, $R^2$ =0.10, $P<0.01$, Fig. 2d), while a negative relationship was found between the microbial N:P ratio and MAT (Topsoil, $R^2$ =0.19, $P<0.001$; Fig. 2f). In discussion: The increase in the microbial C:N ratio and decrease in the microbial N:P ratio that were found along a temperature gradient in this study are in accordance with the findings of Li et al. (2015) and Chen et al. (2016), who reported similar variations in microbial stoichiometry along latitudinal gradients. Temperature drives the variation in the growth of the microbial community, as high growth rates at low latitudes require high RNA contents, causing the N:P ratio to decline (Chadwick et al., 1999; Kooijman et al., 2009; Xu et al., 2013).

Results and Discussion: The SEM showed that climate have indirect effects through AGB, SOC and F:B. These are also interesting results, but was not mentioned in results. I also suggest to added some discussions of these indirect effects, though some

of them were mentioned in discussion implicitly. Anyway, these indirect effects are part of the full picture how climate affects soil microbial stoichiometry. Response: Thanks for your comments. We agree that indirect effects are part of the full picture how climate affects soil microbial stoichiometry. However, the SEM mainly was designed to test the direct effects of potential driving factors. We have revised as follow:"In particular, drought, decreasing aridity index, could affect the growth and productivity of plant, then shape the shift in vegetation types along this grassland transect (Jaleel et al., 2009; Cherwin & Knapp, 2012). "

By the way, the MS used many abbreviations, which markedly decrease the readability. Please try to remove unnecessary ones (e.g. MS, TS, DS in Table 1) ResponseïjŽWe have revised the several abbreviations such as SIC, TC and TP. We also have removed the unnecessary abbreviation like MS, TS, DS in table and figures.

Minor comments: L45: meaning not very clear. Response: We have removed the speculative statements. We have revised the sentence as "The results of this study suggested that the flexibility of the microbial N:P ratio should be considered when establishing the minimum sampling depth in a vertical study for microbial C:N:P stoichiometry study of surface soil."

L58: A few studies. You have listed some studies along latitudinal and environmental gradients in the subsequent text. Response: Thanks for your suggestion. Typo corrected.

L62, 63: with higher latitude? Response: Thank you. Typo corrected.

L64: replace values with patterns Response: Corrected as your suggestion. L142: and at a depth of 10 cm, what does this mean? Response: Thanks for your suggestion. Here we mean that soil depth of 10 cm as surface soil could influence the research on the vertical patterns of microbial stoichiometry. We have revised this sentence as "(iii) to adapt to the imbalance of resources, microbial C:N, C:P and N:P ratios vary between soil depths and at a depth of 10 cm as upper soil, which could influence the

research on the vertical patterns of microbial stoichiometry."

L 171: aridity index (AI) Response: Thank you. Typo corrected.

L217: "were well constrained (Fig. A2)". It needs to be explained. Response: The results indicate well-constrained relationships among C, N and P in soil microbial biomass (Fig. A2). Here we mean that well correlations were found among C, N and P in soil microbial biomass.

L219: larger->higher Response: Thank you. Typo corrected.

L223: the microbial C:N ratio in the subsoil was significantly higher than that in the topsoil (Fig. 2b). This result can not be found in Fig 2b. I guess it was in Table2. Response: We have revised Fig. 2b to Table.2

L238 (and elsewhere) Effects of potential driving factors Response: Thank you. Typo corrected.

L303: microbial C:N:P stoichiometry impacted the microbial community structure as a result of the F:B ratio. Do you mean that C:N:P stoichiometry affects microbial community structure? This seems you be conflict with the SEM. In SEM (and the sentence in line 300-303), the logic is that C:N:P stoichiometry changes as a result of community structure. Response: Thanks for your comments. Given that specific microorganisms (e.g. bacteria and fungi) may have unique elemental compositions, changes in soil microbial communities may lead to differences in the element ratios in biomass (Strickland & Rousk, 2010; Mouginot et al., 2014; Zimmerman et al., 2014). As we shown in SEM, here we mean that the microbial community impacted the microbial C:N:P stoichiometry. We have revised as follows:"An experiment indicated that fungi have lower resource requirements and higher C:N and C:P ratios than bacteria, and thus the microbial community structure impacted the microbial C:N:P stoichiometry as a result of the F:B ratio (Mouginot et al., 2014). "

Fig. 1 Where are the difference among the biomes here? In the caption (also in that of

Table 1), you mentioned: MS, meadow steppe; TS, typical steppe; DS, desert steppe. But you did not show the results at all. Response: Thanks for your reminder! We have removed the speculative statements. This paper mainly focused on the difference between upper soil and lower soil layer, not among the biomes.

Table 2: across 404 the Inner Mongolian grassland at ??? Biome: soil depth? You did not compared biomes here in the Table. Response: Thanks for your reminder! We have revised the Table.2.

Fig. 3: Why the figures of F:B were different from others? It seems many plots have a same F:B value. Response: Thanks for your suggestion. Firstly, as I mentioned in the uncertainties and perspectives, the determination of fungal and bacterial biomasses by PLFA markers, which have limited targets for fungi and bacteria. Secondly, considering relationships to environmental factors, previous studies found that shifts in fungal:bacterial ratio dominance were not always in line with the general expectation. This is likely because the traits expected to differentiate bacteria from fungi are often not distinct (Mouginot et al., 2014). Finally, we analyzed the data to test the reliability of F:B ratio. Compared to researches in similar study area, the result of F:B ratio demonstrated the similar pattern along precipitation gradient. Reference: Chadwick, O.A., Derry, L.A., Vitousek, P.M., Huebert, B.J., Hedin, L.O., 1999. Changing sources of nutrients during four million years of ecosystem development. Nature 397, 491-497. Cherwin, K., & Knapp, A. (2012). Unexpected patterns of sensitivity to drought in three semi-arid grasslands. Oecologia, 169(3), 845-852. Chen, Y. L., Chen, L. Y., Peng, Y. F., Ding, J. Z., Li, F., Yang, G. B., Zhang, B. B. (2016). Linking microbial C:N:P stoichiometry to microbial community and abiotic factors along a 3500‐km grassland transect on the Tibetan Plateau. Global Ecology & Biogeography, 25(12), 1416-1427. Li, P., Yang, Y., Han, W., Fang, J., 2015. Global patterns of soil microbial nitrogen and phosphorus stoichiometry in forest ecosystems. Global Ecology & Biogeography 23, 979-987. Jaleel, C. A., Manivannan, P., Wahid, A., Farooq, M., Al-Juburi, H. J., Somasundaram, R., International Journal of Agriculture& Biology.

(2009). Drought stress in plants: A review on morphological characteristics and pigments composition. 11(1), 100-105. Kooijman, A. M., Mourik, J. M. V., & Schilder, M. L. M. (2009). The relationship between N mineralization or microbial biomass N with micromorphological properties in beech forest soils with different texture and pH. Biology & Fertility of Soils, 45(5), 449. Trabucco, A., and Zomer, R.J. 2009. Global Aridity Index (Global-Aridity) and Global Potential Evapo-Transpiration (Global-PET) Geospatial Database. CGIAR Consortium for Spatial Information. Published online, available from the CGIAR-CSI Geo Portal at: http://www.csi.cgiar.org. Hijmans, R.J., Cameron, S.E., Parra, J.L., Jones, P.G. & Jarvis, A. (2004) The WorldClim interpolated global terrestrial climate surfaces, version 1.3 . Available at http://biogeo.berkeley.edu/. Mouginot, C., Kawamura, R., Matulich, K.L., Berlemont, R., Allison, S.D., Amend, A.S., Martiny, A.C., 2014. Elemental stoichiometry of Fungi and Bacteria strains from grass-land leaf litter. Soil Biology & Biochemistry 76, 278-285. Strickland, M. S., & Rousk, J. (2010). Considering fungal: bacterial dominance in soils – Methods, controls, and ecosystem implications. Soil Biology and Biochemistry, 42(9), 1385-1395. Xu, X., Thornton, P.E., Post, W.M., 2013. A global analysis of soil microbial biomass carbon, nitrogen and phosphorus in terrestrial ecosystems. Global Ecology & Biogeography 22, 737–749. Zimmerman, A. E., Allison, S. D., & Martiny, A. C. (2014). Phylogenetic constraints on elemental stoichiometry and resource allocation in heterotrophic marine bacteria. 16(5), 1398-1410. Zomer, R.J., Trabucco, A., Bossio, D.A, van Straaten, O., Verchot, L.V. 2008. Climate Change Mitigation: A Spatial Analysis of Global Land Suitability for Clean Development Mechanism Afforestation and Reforestation. Agric. Ecosystems and Envir. 126: 67-80.

Please also note the supplement to this comment:
https://www.biogeosciences-discuss.net/bg-2019-351/bg-2019-351-AC1-supplement.pdf

---

## Author Comment (AC2) · 8 Jan 2020

The manuscript requires further clarification on methods, resolution of data and a more realistic presentation of the data analysis. The study design cannot answer the title of the paper, the 'gradient' variables are poorly described, and the methods are lacking with respect to the most important 'variables'. To some extent, the required revisions are minor. However, the focus on 'depth' in the manuscript title suggests that the authors need a major revision with respect to their study hypothesis. Please see the specific comments for further direction on the required revisions.

Response: Thanks for your helpful comments. Then, limited by sampling soil depths, we tended to remove "How deep do we dig for surface soil?" in title. Finally, we have carefully revised our manuscript according to your suggestions. Please see more details in our reply to your specific comments.

Title: you cannot answer the question 'How deep do we dig for surface soil'? Because you did not dig very deep / or a high dig with high incremental accuracy. Two 10 cm samples do not answer the question.

Response: Thanks for your constructive comments. We have deleted "How

deep do we dig for surface soil?" and revised the title as "A comparison of the patterns of microbial C:N:P stoichiometry between topsoil and subsoil along an aridity gradient".

L42 is influenced the correct term? What was the relationship?

Response: Thanks a lot. This sentence has been modified as "The results also revealed that the aridity index (AI) and plant aboveground biomass (AGB) exerted NEGATIVE impacts on the microbial C:N ratio at both soil depths, and the effects of AI decreased in the subsoil."

L87 why 'might be'?

Response: We have removed the speculative statements. We revised the sentence as "Moreover, edaphic variables, such as SOC (Maria et al., 2014; Chen et al., 2016) and soil texture (Li et al., 2015), could be associated with nutrient mineralization and availability, thus influencing the C:N:P stoichiometry in microbial biomass (Griffiths et al., 2012). "

L109 revise wording 'climate change background'. This study does not truly address deeper soils.

Response: Thanks for your suggestions. We revised the sentence as "Such knowledge of the nature of soil microbial stoichiometry is fundamental to understanding ecosystem function, especially within the soil depth of 10-

20cm, which remains uncertain in the published researches."

L132 it this truly an 'ideal' platform. The resolution of the resolution of the aridity index is less than ideal.

Response:Thanks. The sampling sites of this experiment covered meadow steppe, typical steppe and desert steppe, which is a natural environmental gradient. Aridity index ranges from 0.16 to 0.54 along the grassland transect, which offers an ideal experiment platform.

L135 at two depth: why especially in the surface. This is common?

Response: Thanks for your comment. Most studies of soil microbiology have focused exclusively on the soil surface limited to 20 cm in depth, where the densities of microorganisms are highest.

L150 what is the proportion of snow?

Response: Thanks for your suggestion. This sentence has been modified as follows: "The mean annual precipitation (MAP) ranges from 104 to 412 mm, about 80 % of which falls in the growing season from May to August (Chen et al., 2013). "

L158 define slightly? Agricultural? Heavy grazing? Infrequent grazing?

Response: Many thanks for your comments. We defined the slightly

disturbed as the condition that occasional animal bite marks have been observed in our plots, but without agricultural activity or grazing.

L159 why the uneven sample numbers per grassland type? Was this weighted by area?

Response: This study was conducted along natural environment gradient (precipitation, temperature etc.) which shapes the grassland types in this grassland transect. The experiment was designed for comparing the difference between the upper soil and lower soil, not the difference among grassland types.

L161 what stop at 20 cm? L161 where were the three plots sampled? Corners and centre?

Response: Most studies have focused exclusively on the surface 20 cm soil where the densities of microorganisms are highest. However, most studies used 0-10 cm as the surface soil to facilitate sampling and comparative research (Cleveland and Liptzin, 2007; Li and Chen, 2004; Chen et al., 2016). To identify the soil depth that is appropriate for sampling and to improve the understanding of surface soil research at a global scale, we designed a study that divided the surface soil into 0-10 cm and 10-20 cm depths to compare the differences in microbial stoichiometry at the regional scale.

As shown below, there are five 1×1 m$^2$ subplots established at each corner and the center of a 10×10 m$^2$ plot.

[Figure]

L164 sentence is incomplete.

Response: Thanks. We have modified this sentence: "After gentle homogenization and removal of roots, the soil was sieved through a 2-mm mesh and then stored for further experiments. "

L168 what elemental contents? Carbon only? What are the other elements? If other elements, how were they measured?

Response: These is no other element discussed in this paper. The SOC is obtained by subtracting the soil inorganic carbon from the total carbon in this paper.

L170 how was organic matter and carbonates removed from the soil? Carbonates should not be removed before texture is estimated. They are

part of the mineral soil texture.

Response: Carbonate is removed by hydrochloric acid water wash. It is true that carbonates are part of the mineral soil texture. However, the microbial C:N:P stoichiometry was not affected by carbonate. Thanks for your understanding.

L172 what was the resolution of the AI database? Is this adequate to evaluate against site specific measurements? If the metric is important why not calculate at each site?

Response: Thanks for your suggestions. We are sorry that we missed this information. We have included more details on the data extraction and data acquisition methods in the revised manuscript. We have revised it as "Aridity index was extracted them from the Global Aridity Index (Global-Aridity) dataset, which provide high-resolution (30 arc-seconds or ~ 1km at equator) global raster climate data for the 1950-2000 period (http://www.cgiarcsi.org) (Zomer, Trabucco, Bossio, & Verchot, 2008). The specific calculation formula is as follows:

$$Aridity\ Index\ (AI) = MAP\ /\ MAE$$

$$PET=0.0023 \cdot RA \cdot (Tmean+17.8) \cdot TD^{0.5} (mm/month)$$

where MAP represents mean annual precipitation, obtained from the WorldClim Global Climate Data (Hijmans et al. 2005); MAE represents

mean annual potential evapo-transpiration (PET); Tmean represents monthly mean temperature, TD is calculated as the difference between monthly maximum and minimum temperatures; RA represents the extra-terrestrial radiation on top of atmosphere.

L172 what about bulk density? How was it measured? Reported? Why not use loss-on-ignition?

Response: Bulk density with soil volume measured by coating natural clods in cutting ring then weighing the oven-dried clod in drying oven at 105℃ for 24h. Bulk density is calculated by dividing the weight of the oven dried clod by this volume (g·cm$^{-3}$).

How was AGB biomass measured. This is not explained but is an important measure (as indicated by the abstract)

Response: Thanks. We have revised in manuscript as "We measured the aboveground biomass by harvesting the aboveground part of the plants."

L180 is that ration based on mass or volume?

Response: Thanks for your suggestion. We have revised as follow: The

fumigated and nonfumigated samples were extracted using 0.5 M K2SO4 with a soil:solution mass ratio of 1:4.

L193 what different phases?

Response: Thanks for your comment. Here we mean that there are different phases in the process. Phospholipids were separated from neutral and glycolipids on solid-phase extraction columns by eluting with $CHCl_3$, acetone and methanol, respectively. We have revised as follows: "The resultant fatty acid methyl esters were separated, quantified, and identified using capillary gas chromatography."

L201 why t-test? Maybe an ANOVA should be used to account for the different grassland types? Or was a t-test applied to each type? If the latter, was the p value corrected for multiple tests?

Response: It was ture that this study was conducted on three grassland types, and it was also done along the natural environment gradient (e.g. temperature, precipitation, aridity index) in this grassland transect. Owing to our uneven samping, we conducted the correlation analysis to see the change trend along the environment gradient.

L206 AGB is not defined. L206 provide more details on the source of AI and AGB. What is there resolution? Is there a gradient in the data?

Demonstrate that they are gradients. How are they estimated / measured? Provide a description of the data in the results (if they are important variables).

Response: Thanks for your suggestions. We are sorry that we missed this information. We have included more details on the data extraction and data acquisition methods in the revised manuscript. The plant community in subplots was identified, and the above-ground biomass (AGB) was harvested. As to the calculation of AI, we mentioned in the previous reply.

[Figure]

Figure A1. Geographic locations of the sampling sites in the Inner

Mongolian grassland

As Figure A1shown, our sampling sites were distributed along the aridity index gradient. In Inner Mongolia grasslands, the aridity exhibits a gradient that increases from northeast to southwest (aridity index ranges from 0.16 to 0.54). We have added the Figure A1 to the manuscript.

L214 what is the gradient?

Response: As shown in the Figure A1, the aridity exhibits a gradient that increases from northeast to southwest (aridity index ranges from 0.16 to 0.54). Besides, the study area covered both temperature (mean annual temperature ranges from -2.09 to 7.67) and precipitation (mean annual precipitation ranges from 153.9 to 401.7) gradients.

L216 does distinct mean 'significantly different'

Response: We have revised the sentence as "Significantly different water contents, soil bulk density, sand percentages and SOC contents were found between soil depths (P <0.05, Fig. 1a, 1b, 1c, 1f)."

L216 why is bulk density mentioned here: : : and only here? How was it measured? Did it differ greatly between grassland types?

Response: Bulk density is shown here to show the differences in

physicochemical properties between different soil layers. The measurement method was mentioned in the previous reply.

Why was soil microbial biomass not weighted by bulk density?

Response: In the common way, we performed the usual operation instead of weighting by bulk density. Thanks for your understanding!

L218 the concentrations were larger but was the pool larger? Use the bulk density to evaluate the pool difference.

Response: Here we mean that the concentrations of microbial biomass C, N and P, not the pool. We have revised as "The microbial biomass C, N and P concentrations in the topsoil were significantly higher than that in the subsoil (P <0.05, Table. 2)." Thanks for your understanding!

L225-235 these are very weak significant relationships. This should be acknowledged. Similarly, the relationships in Figure 2 and Figure 3 are not very convincing of a relationship(s).

Response: Thanks for your suggestions. We assume that R2 is good enough to exhibit the change trend. Firstly. The variations in microbial C:N and C:P ratios were partly induced by the measurement method. At the small scale, correlations between fumigation-incubation and

fumigation-extraction were variable, which might cause variations in microbial biomass C:N:P stoichiometry (Wardle & Ghani, 1995). Therefore, we assume the variations in microbial biomass C:N:P stoichiometry are inevitable systematic errors.

Second, in a previous study, low R2 also was found along environmental gradients (precipitation, temperature, soil pH, soil content percentage, etc.) at regional scale (Chen et.al 2016). Finally, several global researches only showed the trend but not even R2(Xu et al., 2013; Li et al., 2016). This study offered the regional evidence through measurements across a 2100-km climatic transect in the Inner Mongolian grasslands.

All in all, we do believe the R2 is good enough to exhibit the trend of microbial C:N along the aridity index gradient. We appreciated that you could accept our explanations.

L225-235 you are regression carbon against a ratio that contains carbon: : : this is spurious?

Response: Soil organic matter includes the labile (rapid turnover) and stabilized (slow turnover) fractions (Parton et al, 1987). However, the clear and broad consensus is that soil microbes are primarily limited by C availability (Fierer et al. 2003). There is a clear assumption that available C limits biomass and activity (Eilers et al.2012). As soil carbon matter

changes, microbial biomass N and microbial biomass P change asymmetrically, which affects the ratios (Mooshammer et al. 2014). Therefore, we assume that the regression of SOC against microbial C:N:P stoichiometry is a reasonable analysis.

L237 clarify: : : subsoil is reported in L232, L233, and L235.

Response: Thanks for your helpful comments. We have revised this sentence as: "No or only weak association was found between the microbial C:N, C:P and N:P ratios and the AGB and F:B ratio in the subsoil (Fig. 3)."

L274 drought? Clarify.

Response: Thanks for your helpful comments. As the Table shown, decreasing aridity index means drier weather condition. We have revised the manuscript as follow:"In addition, microbial C:N ratio decreased with decreasing aridity index, which serves as a protective mechanism as microbes decrease their nitrogen use efficiency (NUE, the ratio of N invested in growth over total N uptake) and tend to be more N conservative under dry climatic conditions (Mooshammer et al., 2014; Delgado-Baquerizo et al., 2017)." Due to the weak relationship, we have mentioned in the previous reply.

**Table.** Generalized climate classification scheme for *Global*-Aridity values (UNEP

1997).

| Aridity Index Value | Climate Class |
|---|---|
| < 0.03 | Hyper Arid |
| 0.03 – 0.2 | Arid |
| 0.2 – 0.5 | Semi-Arid |
| 0.5 – 0.65 | Dry sub-humid |
| > 0.65 | Humid |

L285 many things change across latitude. Is microbial biomass influenced by latitude or the change in grassland type / climate / etc. Will microbial biomass also change across longitude? What is the range in the aridity gradient in the current study?

Response: In general, latitude pattern was driven by temperature. Therefore, we have added results of the temperature analysis to make our conclusions more robust. It is true that microbial biomass exhibits longitudinal pattern in global study (Xu et al., 2013). As shown in Figure A1 and Table 1, aridity index ranges from 0.16 to 0.54 in this study.

L292 this is essentially stating that carbon is related to a ratio that includes carbon. This is not surprising. Is this a spurious (correlation) regression?

Response: Soil organic matter includes the labile (rapid turnover) and stabilized (slow turnover) fractions (Parton et al, 1987). However, the clear and broad consensus is that soil microbes are primarily limited by C availability (Fierer et al., 2003). There is a clear assumption that available

C limits biomass and activity (Eilers et al., 2012). As soil carbon matter changes, microbial biomass N and microbial biomass P change asymmetrically, which affects the ratios (Mooshammer et al.,2014). Therefore, we assume that the correlation regression with SOC is a reasonable analysis.

L309 how were AGB and AI measured? Are they site specific or regional indicators? They only show a weak relationship with little predicative power.

Response: Thanks a lot. The above ground biomass was site-specific while aridity index was a regional indicator. We measured the aboveground biomass by harvesting the aboveground part of the plants. As to the source of AI, more details in the previous reply.

L334 did you quantify spatial heterogeneity? How?

Thanks for your comments. We have removed the speculative statements. The highly variable N:P ratio in microbes may reflect the high variability in site-related P availability (Chen et al. 2013; Li, et al. 2015). Furthermore, the relatively high microbial N:P ratio (suggesting P limitation) are supported by direct evidence showing that low soil P availability strongly limits microbial biomass, activity, and other ecosystem processes (Cleveland et al., 2007). This sentence has been modified as "The high variability of the N:P ratio in soil and soil microbial biomass therefore

indicates that the N:P ratio could be an indicator of the ecosystem nutrient status at deeper soil depths (Cleveland et al., 2007; Chen et al. 2013; Li, et al. 2015)."

L337 you cannot answer this question.

Response: Thanks for your comment. We agree that inappropriate statement might result in uncertainty. This sentence has been modified as follows: "How deep should we dig to evaluate the surface soil the microbial stoichiometry in vertical study?"

L341 are the pools distinct?

Response: Thanks a lot. We have removed the speculative statements. We have revised as follows: The results showed significant differences in the water content and sand percentage, SOC content and F:B ratio between soil depths, suggesting that the resource supplies between topsoil and subsoil were significantly different.

L350 you tested limited depth, with course increments.

Response: Similar findings were reported in the top 16 cm of soil in a Mediterranean oak forest (0-8 cm and 8-16 cm), where the microbial nutrient ratios (C:N, C:P and N:P) varied between soil depth (Aponte et al., 2010).

L369 what about pools?

Response: Thanks a lot. This study focused on the C:N, C:P and N:P ratios in microbial biomass, not the pools of microbial biomass C, N, P. We don't think that's an important variable.

L375 not shown, this statement is too strongly with respect to drought. There was a weak relationship using a coarse metric.

Response: In addition, microbial C:N ratio decreased with decreasing aridity index, consistent with the perspective that microbes mediate their nitrogen use efficiency and tend to be more N conservative under drier climatic conditions. In terms of the weak relationship, we have mentioned in the previous reply.

L383 edaphic? Influence of soil on soil?

Response: Edaphic factor means any characteristic of the environment resulting from the physical, chemical or biotic components of the soil such as the microbial structure, soil texture and soil organic content. In our results, the microbial C:N, C:P and N:P ratios were influenced by SOC and F:B ratio.

L384 you need to demonstrate the gradient

Response: As the most important gradient, AI gradient is demonstrated in Figure A1. Table 1 also showed the ranges of mean annual temperature, mean annual precipitation and above ground biomass in this study.

Reference:

Chen, D., J. Cheng, P. Chu, S. Hu, Y. Xie, I. Tuvshintogtokh & Y. Bai (2015) Regional-scale patterns of soil microbes and nematodes across grasslands on the Mongolian plateau: relationships with climate, soil, and plants. Ecography, 38, 622-631.

Chen, Y., W. Han, L. Tang, Z. Tang & J. Fang (2013) Leaf nitrogen   n and phosphorus concentrations of woody plants differ in responses to climate, soil and plant growth form. 36, 178-184.

Chen, Y. L., Chen, L. Y., Peng, Y. F., Ding, J. Z., Li, F., Yang, G. B., Zhang, B. B. (2016). Linking microbial C:N:P stoichiometry to microbial community and abiotic factors along a 3500-km grassland transect on the Tibetan Plateau. Global Ecology & Biogeography, 25(12), 1416-1427.

Cleveland, C. C., and Liptzin, D. 2007: C:N:P Stoichiometry in Soil: Is There a "Redfield Ratio" for the Microbial Biomass?, Biogeochemistry, 85, 235-252, 2007.

Delgado-Baquerizo, M., Powell, J.R., Hamonts, K., Reith, F., Mele, P., Brown, M.V., Dennis, P.G., Ferrari, B.C., Fitzgerald, A., Young, A., 2017. Circular linkages between soil biodiversity, fertilityand plant productivity are limited to topsoil at the continental scale. New Phytologist 215.

Eilers, K. G., Debenport, S., Anderson, S., & Fierer, N. (2012). Digging deeper to find unique microbial communities: The strong effect of depth on the structure of bacterial and archaeal communities in soil. Soil Biology and Biochemistry, 50, 58-65.

Li, P., Yang, Y., Han, W., Fang, J., 2015. Global patterns of soil microbial nitrogen and phosphorus stoichiometry in forest ecosystems. Global Ecology & Biogeography 23, 979-987.

Li, X. Z., and Z. Z. Chen, 2004. Soil microbial biomass C and N along a climatic transect in the Mongolian steppe. Biology & Fertility of Soils 39(5):344-351.

Mooshammer, M., Wanek, W., Hämmerle, I., Fuchslueger, L., Hofhansl, F., Knoltsch, A., Schnecker, J.,Takriti, M., Watzka, M., Wild, B., 2014. Adjustment of microbial nitrogen use efficiency to carbon:nitrogen imbalances regulates soil nitrogen cycling.

5, 3694.

Wardle, D. A., & Ghani, A. (1995). Why is the strength of relationships between pairs of methods for estimating soil microbial biomass often so variable? Soil Biology and Biochemistry, 27(6), 821-828.

Trabucco, A., and Zomer, R.J. 2009. *Global Aridity Index (Global-Aridity) and Global Potential Evapo-Transpiration (Global-PET) Geospatial Database*. CGIAR Consortium for Spatial Information. Published online, available from the CGIAR-CSI GeoPortal at: http://www.csi.cgiar.org.

UNEP (1997) World atlas of desertification. United Nations Environment Programme

Parton, W. J., D. S. Schimel, C. V. Cole, and D. S. Ojima. 1987. Analysis of Factors Controlling Soil Organic Matter Levels in Great Plains Grasslands1.Soil Science Society of America Journal J. 51:1173-1179.

Fierer, N., Schimel, J. P., & Holden, P. A. (2003). Variations in microbial community composition through two soil depth profiles. Soil Biology & Biochemistry, 35(1), 167-176.

Xu, X., Thornton, P. E., and Post, W. M.: A global analysis of soil microbial biomass carbon, nitrogen and phosphorus in terrestrial ecosystems, Global Ecology & Biogeography, 22, 737–749, 2013.

---

## Author Response (AR2)

General comments: This manuscript (MS) examined soil microbial C:N:P stoichiometry along a large aridity gradient across different temperate grassland biomes, using 58 plots sampled from a 2100-km transect in Inner Mongolian, China. The dataset is good not only in that studies of soil microbial stoichiometry along great aridity gradient is still limited, and also because they examined difference in patterns and potential drivers between top and subsoil. The MS is generally well written, though there were some clear typewriting flaws and some sentences not easy to understand. I suggest a minor revision, mainly on the improving of the statistical analyses, and clarity and readability of the MS.

Response: Thanks for your positive comments. We have carefully revised our manuscript according to your suggestions. Please see more details in our response to your specific comments.

Specific comments: The title: from the title I have first thought that you have sampled much deeper than 10 cm. However, I then realized that you have sampled to a depth of 20 cm. I would suggest to revise the title and delete "How deep do we dig for surface soil?". The rest part of the title is good enough.

Response: Thanks for your constructive comments. Following your

suggestion, we have revised title as "A comparison of patterns of microbial

C : N : P stoichiometry between topsoil and subsoil along an aridity

gradient".

Methods: In 2.2, how the above-ground biomass data was obtained was not

clarified, but this data was used in statistical analyses. In addition, the method to

calculate aridity index need to be introduced. Though the data was extracted from

database, the Equation needs to be introduced for readers to better understand the

biological meanings. Further, there were several different indices for aridity.

Response: Thanks for your suggestions. We are sorry that we missed this

information. We have included more details on the methods of data

extracting in the revised manuscript. We measured the aboveground

biomass by harvesting the aboveground part of the plants in the sampling

plot. We have revised it as "The aridity index was extracted from the Global

Aridity Index (Global-Aridity) dataset,which provides high-resolution (30

arc-seconds or ~ 1km at the equator) global raster climate data for the

period 1950-2000 (http://www.cgiarcsi.org) (Zomer, Trabucco, Bossio, &

Verchot, 2008). The specific calculation formula is as follows:

$$Aridity\ Index\ (AI) = MAP / MAE$$

$$PET=0.0023·RA·(Tmean+17.8)·TD0.5(mm/month)$$

where MAP represents the mean annual precipitation, obtained from the

WorldClim Global Climate Data (Hijmans et al. 2005); MAE represents the

mean annual potential evapo-transpiration (PET); Tmean represents the monthly

mean temperature, TD is calculated as the difference between the monthly

maximum and minimum temperatures; and RA represents the extra-terrestrial radiation on above the atmosphere.

Statistical analyses: The plots were sampled from a northeast to southwest transect, which include variations in both temperature and aridity. It remains unclear how temperature contribute to the geographic patterns reported here. Considering the large difference in aridity from typical steppe to desert steppe, personally I agree with your results on the role of aridity on microbial stoichiometry. However, you may consider to include temperature as a predictor, to make your conclusions more robust.

Response: thank you for your helpful suggestions. Indeed,I consider temperature as an important indicator. We have added  related figure, result and discussion in the manuscript.

[Figure]

Figure 2. Relationships between the C:N, C:P and N:P ratios in soil microbial biomass and aridity index (a-c), mean annual temperature (d-e) and latitude (g-i) in the Inner Mongolian grassland.

In result: In addition, the decreasing trend was found between the microbial C:N ratio and MAT (Topsoil, $R^2 = 0.14$, P< 0.01; Subsoil, $R^2 = 0.10$, P< 0.01, Fig. 2d), while a significant negative relationship was found between the microbial N:P ratio and MAT (Topsoil, $R^2 = 0.19$, P< 0.001; Fig. 2f).

In discussion: The increase in the microbial C:N ratio and the decrease in the microbial N:P ratio that were found along a temperature gradient in this

study are in accordance with the findings of Li et al. (2015) and Chen et al. (2016), who reported similar variations in microbial stoichiometry along latitudinal gradient. Temperature drives the variation in the growth of the microbial community, as high growth rates at low latitudes require high RNA contents, causing the N:P ratio to decline (Chadwick et al., 1999; Kooijman et al., 2009; Xu et al., 2013).

Results and Discussion: The SEM showed that climate have indirect effects through AGB, SOC and F:B. These are also interesting results, but was not mentioned in results. I also suggest to added some discussions of these indirect effects, though some of them were mentioned in discussion implicitly. Anyway, these indirect effects are part of the full picture how climate affects soil microbial stoichiometry.

Response: Thanks for your comments. We agree that indirect effects are part of the full picture how climate affects soil microbial stoichiometry. However, the SEM mainly was designed to test the direct effects of potential driving factors. We have revised as follow:" In particular, drier weather condition, and the decreasing aridity index, could affect the growth and productivity of plants, and then shape a shift in vegetation types along this grassland transect (Jaleel et al., 2009; Cherwin & Knapp, 2012)."

By the way, the MS used many abbreviations, which markedly decrease

the readability. Please try to remove unnecessary ones (e.g. MS, TS, DS in Table 1)

Response: We have revised the several abbreviations such as SIC, TC and TP. We also have removed the unnecessary abbreviation like MS, TS, DS in table and figures.

Minor comments: L45: meaning not very clear.

Response: We have removed the speculative statements. We have revised the sentence as "The results of this study suggested that the flexibility of the microbial N:P ratio should be considered when establishing the sampling depth for microbial stoichiometry study in topsoil."

L58: A few studies. You have listed some studies along latitudinal and environmental gradients in the subsequent text.

Response: Thanks for your suggestion. Typo corrected.

L62, 63: with higher latitude?

Response: Thank you. Typo corrected.

L64: replace values with patterns

Response: Corrected as your suggestion.

L142: and at a depth of 10 cm, what does this mean?

Response: Thanks for your suggestion. Here we mean that soil depth of 10 cm as surface soil could influence the research on the vertical patterns of microbial stoichiometry. We have revised this sentence as " In addition, the identification of soil depth for vertical study is differernt in some published literature (Li and Chen, 2004; Aponte et al., 2010; Tischer et al., 2014; Peng and Wang, 2016). We predicted that variation of bacterial and fungal taxa between soil depths might contribute to the shifts in C:N:P stoichiometry, especially in the N:P ratio (Mouginot et al., 2014; Camenzind et al., 2018). Therefore, we focus on (i) the effects of potential driving factors on microbial C:N, C:P and N:P ratios in topsoil and subsoil (ii) the response of the microbial C:N, C:P and N:P ratios to soil depth."

L171: aridity index (AI)

Response: Thank you. Typo corrected.

L217: "were well constrained (Fig. A2)". It needs to be explained.

Response: The results indicate well-constrained relationships among C, N and P in soil microbial biomass (Fig. A2). Here we mean that well correlations were found among C, N and P in soil microbial biomass.

L219: larger->higher

Response: Thank you. Typo corrected.

L223: the microbial C:N ratio in the subsoil was significantly higher than that in the topsoil (Fig. 2b). This result can not be found in Fig 2b. I guess it was in Table2.

Response: We have revised Fig. 2b to Table.2

L238 (and elsewhere) Effects of potential driving factors

Response: Thank you. Typo corrected.

L303: microbial C:N:P stoichiometry impacted the microbial community structure as a result of the F:B ratio. Do you mean that C:N:P stoichiometry affects microbial community structure? This seems you be conflict with the SEM. In SEM (and the sentence in line 300-303), the logic is that C:N:P stoichiometry changes as a result of community structure.

Response: Thanks for your comments. Given that specific microorganisms (e.g. bacteria and fungi) may have unique elemental compositions, changes in soil microbial communities may lead to differences in the element ratios in biomass (Strickland & Rousk, 2010; Mouginot et al., 2014; Zimmerman et al., 2014). As we shown in SEM, here we mean that the microbial community impacted the microbial C:N:P stoichiometry. We have revised as follows:" An experiment indicated that fungi have lower resource requirements and higher C:N and C:P

ratios than bacteria; thus, the shift in the F:B ratio impacted microbial C:N:P stoichiometry (Mouginot et al., 2014)."

Fig. 1 Where are the difference among the biomes here? In the caption (also in that of Table 1), you mentioned: MS, meadow steppe; TS, typical steppe; DS, desert steppe. But you did not show the results at all.

Response: Thanks for your reminder! We have removed the speculative statements. This paper mainly focused on the difference between topsoil and subsoil layer, not among the biomes.

Table 2: across 404 the Inner Mongolian grassland at ??? Biome: soil depth? You did not compared biomes here in the Table.

Response: Thanks for your reminder! We have revised the Table.2.

Fig. 3: Why the figures of F:B were different from others? It seems many plots have a same F:B value.

Response: Thanks for your suggestion. Firstly, as I mentioned in the uncertainties and perspectives, the determination of fungal and bacterial biomasses by PLFA markers, which have limited targets for fungi and bacteria.

Secondly, considering relationships to environmental factors, previous studies found that shifts in fungal:bacterial ratio dominance were not always in line with the general expectation. This is likely because the traits

expected to differentiate bacteria from fungi are often not distinct (Mouginot et al., 2014).

Finally, we analyzed the data to test the reliability of F:B ratio. Compared to researches in similar study area, the result of F:B ratio demonstrated the similar pattern along precipitation gradient.

Anonymous Referee #2

How deep do we dig for surface soil? A comparison of patterns of microbial C:N:P stoichiometry between topsoil and subsoil along an aridity gradient

**General comments**

This paper focus on the sampling depth for analysing microbial $C:N:P$ stoichiometry at 0-10 cm or 10-20 cm soil depth. It is an interesting study made in permanent grassland with an aridity index, but the interpretation and presentation of the data should be improved.

The paper will profit from clearer hypothesis that can be tested, and more clear wording and presenting of the results. I do also suggest putting the correlation analyses given in 3.1 into a table, which would make it more accessible for the reader. Your question: "How deep do we dig for surface soil?" Should be clearly answered in the conclusion.

Response: Thanks for your positive comments. We perfer to demonstrate the correlation analyses by figure, which would be more direct in our view. Then, limited by sampling soil depths, we tended to remove "How deep do we dig for surface soil?" in title. Additionally, we have carefully revised our manuscript according to your suggestions. Please see more details in our response to your specific comments.

**Specific comments**

Normally subsoil is used for the soil under the surface soil/ topsoil that are less affected by plant roots and tillage operations. However, I assume there were no tillage at the sites referred to in the present paper. The root distribution and rooting depth for the different sites are not given, but in permanent grassland most of the rooting and microbial activity is in the upper soil layer. I would still be reluctant to use the "subsoil" as a term for the soil layer at 10-20 cm depth as the roots would likely go deeper than 10 cm. Surface soil and topsoil are in many cases used as synonyms and the heading is therefore confusing. I suggest in stead: How deep do

we dig for surface soil? A comparison of patterns of microbial $C:N:P$ stoichiometry between an upper and lower soil layer along an aridity gradient.

Response: Thanks for your suggestions. This study is based on previous research conducted in China's grassland which found the largest proportion of roots near the soil surface (0-30cm). The title has been modified as follows "A comparison of patterns of microbial $C:N:P$ stoichiometry between topsoil and subsoil along an aridity gradient."

When you present hypotheses, it should be possible to test them and to either confirm or reject them and the result of the testing of the hypotheses should be clearly presented in the conclusion.

(i) microbial C:N and C:P ratios increase and the microbial N:P ratio decreases across an aridity gradient because of differences in nutrient-use efficiency. The first part of this hypotheses "microbial C:N and C:P ratios increase and the microbial N:P ratio decreases across an aridity gradient», you have actually tested in the present paper, but the result is not clearly written in the conclusions. In Figure 2, C:N, C:P and N:P ratios are given along an aridity Index (Gradient). Because of the very low relationships between the ratios and the aridity index, this part of the hyphothesis cannot be confirmed. $R^2=0.1$ is very low. In discussion you write: "microbial C:N and C:P ratios increase and the microbial N:P ratio decreases across an aridity gradient»I do not agree with this statement. Because of the low $R^2$, a $P<0.05$ does not say much. If you look at figure 2, you see that the variation in within C:N and C:P sites at the same aridity is much larger than the impact of Aridity index. I would rather call it a trend, then to state a significantly impact.

Response: Thanks for your suggestions. We assume that $R^2$ is good enough to exhibit the change trend. Firstly. The variations in microbial C:N and C:P ratios were partly induced by the measurement method. At the small scale, correlations between fumigation-incubation and fumigation-extraction were variable, which might cause variations in microbial C:N:P stoichiometry (Wardle & Ghani, 1995). Therefore, we assume the variations in microbial biomass C:N:P stoichiometry are inevitable systematic errors.

Second, in a previous study, low $R^2$ also was found along environmental gradients (precipitation, temperature, soil pH, soil content percentage, etc.) at regional scale (Chen et.al 2016). Finally, several global researches only showed the trend but not even $R^2$ (Xu et al., 2013; Li et al., 2016). This study offered the regional evidence through measurements across a 2100-km climatic transect in the Inner Mongolian grasslands.

All in all, we do believe the $R^2$ is good enough to exhibit the trend of microbial C:N along the aridity index gradient. We appreciated that you could accept our explanations.

The second part of the first hypothesis "because of differences in nutrient-use efficiency.», you do only discuss and do not test. I would leave that out from the hypothesis.

Response: Thanks for your helpful suggestion. We have removed the speculative statement.

(ii)   Due to variations in resource supply among different soil depths, the effects of driving factors on microbial C:N, C:P and N:P ratios might decrease with soil depth.

This hypothesis you have not tested and cannot do, as you do not know if variations in resource supply among different soil depths actually do effect driving factors on microbial C:N, C:P and N:P ratios. What you can test is:"Microbial C:N, C:P and N:P ratios do vary with soil depth." In the results 3.1 lines 222 to 223 you write: "Moreover, the microbial C:N ratio in the subsoil was significantly higher than that in the topsoil (Fig. 2b)." I assume you must mean table 2? If this is the case, such a hypothesis could be confirmed for C:N ratio, and rejected for C:P and N:P ratios. Obs,. You write in the abstract (line 32-34) :" We found that the microbial C:N , C:P and N:P ratios varied with soil depth.» According to table 2, they do not.

Response:Thanks for your reminder. Firstly, according to the Table.2, the result showed that the microbial C:N ratio in the lower soil was significantly higher than that in the upper soil. We have revised the error in the manuscript. Secondly, our hypothesis is based on the homeostasis theory and previous literature. Under the framework of homeostasis theory, microorganisms are constrained by basic metabolic needs, which results in microorganisms having fixed C:N:P ratios (Sterner and Elser, 2002). To adapt to the resource imbalances and limitations caused by substrate heterogeneity, microbes exhibit stoichiometric non-homeostasis by regulating their ecological processes such as mineralization and immobilization (Fanin et al.,

2013; Mooshammer et al.,2014). As the depth of the soil changes, a shift in resource supply might lead to a variation in microbial stoichiometry. Published studies also show the variation in microbial C:N, C:P and N:P ratios between soil depths (Aponte et al., 2010; Tischer et al., 2014).Therefore, we assumed that microbial C:N, C:P and N:P ratios do vary with soil depth. We have revised this sentence as "From the topsoil to the subsoil, the microbial C:N, C:P and N:P ratios varied from 6.59 to 6.83, from 60.2 to 60.5 and from 9.29 to 8.91, respectively. Only that the microbial C:N ratio significantly increased with soil depth."

According your suggestion, we have revised this section related to the hypothesis as " In addition, the identification of soil depth for vertical study is differernt in some published literature (Li and Chen, 2004; Aponte et al., 2010; Tischer et al., 2014; Peng and Wang, 2016). We predicted that variation of bacterial and fungal taxa between soil depths might contribute to the shifts in C:N:P stoichiometry, especially in the N:P ratio ((Mouginot et al., 2014; Camenzind et al., 2018). Therefore, we focus on (i) the effects of potential driving factors on microbial C:N, C:P and N:P ratios in topsoil and subsoil (ii) the response of the microbial C:N, C:P and N:P ratios to soil depth."

to adapt to the imbalance of resources, microbial C:N, C:P and N:P ratios vary between soil depths and at a depth of 10 cm, which could influence the research on the vertical patterns of microbial stoichiometry.

I do not understand what you mean by this hypothesis. You should convert it to a hypothesis that can be tested and clearly present the result of the hypothesis. Do you mean "Microbial C:N, C:P and N:P ratios do vary with soil depth. At 0-10 cm depth the ratios are more influenced of an aridity gradient and other ecological factors than at 10-20 cm soil depths"?

Response: Thanks for your suggestion. At the regional scale, the potential influencing factors of microbial stoichiometry were affected by soil depth (Chen et al., 2015; Chen et al., 2016), which will help us to assess the impact of soil depth on soil microbial stoichiometry. Here we mean that soil microbes may shift their C:N:P stoichiometry to adapt to the imbalance of resource between topsoil and subsoil. We have revised section related to the hypothesis accordingly.

In 3.1 you refer to "environmental gradient» in the title, but you do not refer to what you mean with «environmental gradient.»

Response: We majorly foucused on the aridity gradient as our environmental gradient, which combined the effects of temperature and precipitaition.

You do focus on the impact of Latitude, but I do not understand for which purpose. And again the degree of explanation is low ($R^2$= 0.14) and the variation is large.

Response: Thanks for your suggestion. Until now, there are controversial latitudinal patterns of microbial C:N, C:P and N:P ratios. This study offered the regional evidence through measured data across a 2100-km climatic transect in the Inner Mongolian grasslands. we have added analysis related to the mean annual temperature, to make our conclusions more robust. As we mentioned

above, we do believe the $R^2$ is good enough to exhibit the trend of microbial C:N along environment gradient.

Because this study is done on three grassland types (meadow steppe, typical steppe and desert steppe) with corresponding soil types, I do miss the discussion on impact of grassland types, plant roots and rooting pattern on the microbial stoichiometry.

Response: Thanks for your suggestion. First, this study is based on previous research conducted in China's grassland which found the largest proportion of roots near the soil surface (0-30cm). From our previous survey, above ground biomass was nearly proportional to below ground biomass with a scaling exponent across various grassland types in China's grassland. Therefore, above-ground biomass was chosen as important indicator in the SEM.

Secondly, it was ture that this study was conducted on three grassland types and it also was done along natural environment gradient (e.g. temperature, precipitation, aridity index) in this grassland transect. Owing to our uneven samping, we conducted the correlation analysis to see the change trend along environment gradient.

Because aridity gradient (index) is central in this study it should be given how it was calculated (Line 171-172).

Response: Thanks for your suggestion. We are sorry that we missed this information. We have included more details on the methods of data extracting and data acquiring in the revised manuscript. We have revised as "The aridity index was extracted from the Global Aridity Index (Global-

Aridity) dataset,which provides high-resolution (30 arc-seconds or ~ 1km at the equator) global raster climate data for the period 1950-2000 (http://www.cgiarcsi.org) (Zomer, Trabucco, Bossio, & Verchot, 2008). The specific calculation formula is as follows:

$$Aridity\ Index\ (AI) = MAP / MAE$$

$$PET=0.0023 \cdot RA \cdot (Tmean+17.8) \cdot TD0.5(mm/month)$$

where MAP represents the mean annual precipitation, obtained from the WorldClim Global Climate Data (Hijmans et al. 2005); MAE represents the mean annual potential evapo-transpiration (PET); Tmean represents the monthly mean temperature, TD is calculated as the difference between the monthly maximum and minimum temperatures; and RA represents the extra-terrestrial radiation on above the atmosphere."

Figure A3 need some introduction. How did you develop this?

Response: Thanks for your suggestions. We showed the direct pathway and related introduction in Figure. A3. We have revised the table as follows:

| Pathway | Interpretation | Reference |
|---------|----------------|-----------|
| SOC→Microbial C:N:P | Influence of SOC on microbial C:N:P stoichiometry | (Hartman et al., 2013; Maria et al., 2014; Mooshammer et al.,2014) |
| AGB→Microbial C:N:P | Plant necromass represents the fundamental resource for microbes to maintain element balance | (Cleveland et al., 2007; Aponte et al., 2010; Manzoni et al., 2010; Li et al., 2012; Zechmeister-Boltenstern et al., 2016) |
| AI→Microbial C:N:P | Influence of increasing temperature on microbial C and N cycle | (Wang et al., 2014; Zechmeister-Boltenstern et al., 2016; Chen et al., 2016) |

| | | |
|---|---|---|
| Sand percentage→Microbial C:N:P | Influence of soil texture associated water-holding capacity and nutrient availability on microbial C:N:P ratios | (Cleveland et al., 2007; Xu et al., 2013; Maria et al., 2014; Li et al., 2015; Zechmeister-Boltenstern et al., 2016) |
| F:B ratio→ Microbial C:N:P | Influence of a shift in the composition of microbial community on microbial C:N:P ratios | (Ross et al.,1993; Cleveland et al.,2007; Aponte et al., 2010; Tischer et al., 2014; Zechmeister-Boltenstern et al., 2016; Chen et al., 2016) |

**Technical corrections**

Line 181 and line 189, You must explain what a universal conversion factor is, what the units are and give a reference to where you got it from.

Response: Thanks for your suggestions. 0.45 is the conversion factor ($k_{EC}$), no units (Jenkinson et al., 1976). The specific calculation formula is as follows:

$$k_{EC} = EC/ FC$$

where EC represents the difference between organic C extracted by 0.5 M $K_2SO_4$ from fumigated and non-fumigated soil, fumigation-extraction method. FC represents the flush of $CO_2$-C caused by fumigation during a 10 day incubation, fumigation-incubation method (Jenkinson et al., 1976; Vance et al., 1987; Wu and Joergensen et al., 1990; Joergensen et al., 1996).

Line 185 Which principal method is used? Cloroform fumigation? Hedley and Stewart (1982) is not given in the reference list.

Response: Both methods were calculated by the difference in total microbial-P content before and after $CHCl_3$ fumigation. For microbial biomass P, calculation

was based on the difference between P removed by NaHCO$_3$ extraction of CHCl$_3$ fumigated and nonfumigated samples. We have added the source of the method as follows: Hedley, M. J., & Stewart, J. W. B. (1982). Method to measure microbial phosphate in soils. Soil Biology and Biochemistry, 14(4), 377-385.

Line 201 Was the log10 transformed ratios normally distributed?

Response: Thanks for your suggestion. The log10 transformed microbial C:N, C:P and N:P ratios in both soil depths demonstrated normal distribute. We have added the Figure A3 in the manuscipt.

Figure A3. Histograms showing the frequency distributions of the soil microbial C:N, C:P and N:P ratios in the topsoil (a-c) and the subsoil (d-f).

[Figure]

Line 223 , Should it be table 2, not Fig. 2b?

Response: Thanks for your suggestion. Here should be Table.2. We have revised in the manuscript.
The manuscript requires further clarification on methods, resolution of data and a more realistic presentation of the data analysis. The study design cannot answer the title of the paper, the 'gradient' variables are poorly described, and the methods are lacking with respect to the most important 'variables'. To some extent, the required revisions are minor. However, the focus on 'depth' in the manuscript title suggests that the authors need a major revision with respect to their study hypothesis. Please see the specific comments for further direction on the required revisions.

Response: Thanks for your helpful comments. Then, limited by sampling soil depths, we tended to remove "How deep do we dig for surface soil?" in title. Finally, we have carefully revised our manuscript according to your suggestions. Please see more details in our reponse to your specific comments.

Title: you cannot answer the question 'How deep do we dig for surface soil'? Because you did not dig very deep / or a high dig with high incremental accuracy. Two 10 cm samples do not answer the question.

Response: Thanks for your constructive comments. We have deleted "How deep do we dig for surface soil?" and revised the title as "A comparison of the patterns of

microbial C:N:P stoichiometry between topsoil and subsoil along an aridity gradient".

L42 is influenced the correct term? What was the relationship?

Response: Thanks a lot. This sentence has been modified as "The results also revealed that the aridity index (AI) and plant aboveground biomass (AGB) exerted negative impacts on the microbial C:N ratio at both soil depths, and the effects of AI decreased in the subsoil."

L87 why 'might be'?

Response: We have removed the speculative statements. We revised the sentence as " Moreover, edaphic variables, such as SOC contnet (Maria et al., 2014; Chen et al., 2016) and soil texture (Li et al., 2015), could be associated with nutrient mineralization and availability, thus influencing the C:N:P stoichiometry in microbial biomass (Griffiths et al., 2012). "

L109 revise wording 'climate change background'. This study does not truly address deeper soils.

Response: Thanks for your suggestions. Aridity, which is increasing worldwide because of climate change, affects the structure and functioning of dryland ecosystems. We revised the sentence as "Such knowledge of the nature of soil microbial stoichiometry is fundamental for understanding ecosystem function, especially at the 10-20 cm soil depth, which remains highly uncertain in the published studies."

L132 it this truly an 'ideal' platform. The resolution of the resolution of the aridity index is less than ideal.

Response: Thanks. The sampling sites of this experiment covered meadow steppe, typical steppe and desert steppe, which is a natural environmental gradient. Aridity index ranges from 0.16 to 0.54 along the grassland transect, which offers an ideal experiment platform.

L135 at two depth: why especially in the surface. This is common?

Response: Thanks for your comment. Most studies of soil microbiology have focused exclusively on the soil surface limited to 20 cm in depth, where the densities of microorganisms are highest.

L150 what is the proportion of snow?

Response: Thanks for your suggestion. This sentence has been modified as follows: "From northeast to southwest, the mean annual temperature increases from -1.7 to 7.7°C, and the mean annual precipitation decreases from 402 mm to 154 mm, approximately 80% of which falls in the growing season from May to August (Chen et al., 2013). "

L158 define slightly? Agricultural? Heavy grazing? Infrequent grazing?

Response: Many thanks for your comments. We defined the slightly disturbed

as the condition that occasional animal bite marks have been observed in our plots, but without agricultural activity or grazing.

L159 why the uneven sample numbers per grassland type? Was this weighted by area?

Response: This study was conducted along natural environment gradient (precipitation, temperature etc.) which shapes the grassland types in this grassland transect. The experiment was designed for comparing the difference between the topsoil and subsoil, not the difference among grassland types. The sample numbers of grassland type was weighted by area.

L161 what stop at 20 cm? L161 where were the three plots sampled? Corners and centre?

Response: Most studies have focused exclusively on the top 20 cm soil where the densities of microorganisms are highest. However, most studies used 0-10 cm as the topsoil to facilitate sampling and comparative research (Cleveland and Liptzin, 2007; Li and Chen, 2004; Chen et al., 2016). To identify the soil depth that is appropriate for sampling and to improve the understanding of topsoil research at a global scale, we designed a study that divided the topsoil into 0-10 cm and 10-20 cm depths to compare the differences in microbial stoichiometry at the regional scale.

As shown below, there are five 1×1 m² subplots established at each corner and the center of a 10×10 m² plot.

[Figure]

L164 sentence is incomplete.

Response: Thanks. We have modified this sentence: "After gentle homogenization and removal of roots, the soil was sieved through a 2-mm mesh and then stored for further experiments."

L168 what elemental contents? Carbon only? What are the other elements?
If other elements, how were they measured?

Response: We measured the content of soil organic matter and the content of C, N, P in microbial biomass. The SOC contnet is obtained by subtracting the soil inorganic carbon from the total carbon in this paper.

L170 how was organic matter and carbonates removed from the soil?

Carbonates should not be removed before texture is estimated. They are

part of the mineral soil texture.

Response: The carbonate is removed by hydrochloric acid water wash. It is true that carbonates are part of the mineral soil texture. However, the microbial C:N:P stoichiometry was not affected by carbonate. Thanks for your understanding.

L172 what was the resolution of the AI database? Is this adequate to evaluate against site specific measurements? If the metric is important why not calculate at each site?

Response: Thanks for your suggestions. We are sorry that we missed this information. We have included more details on the data extraction and data acquisition methods in the revised manuscript. We have revised it as "The aridity index was extracted from the Global Aridity Index (Global-Aridity) dataset,which provides high-resolution (30 arc-seconds or ~ 1km at the equator) global raster climate data for the period 1950-2000 (http://www.cgiarcsi.org) (Zomer, Trabucco, Bossio, & Verchot, 2008). The specific calculation formula is as follows:

$$\text{Aridity Index (AI)} = \text{MAP} / \text{MAE}$$

$$\text{PET} = 0.0023 \cdot \text{RA} \cdot (\text{Tmean} + 17.8) \cdot \text{TD}0.5 (\text{mm/month})$$

where MAP represents the mean annual precipitation, obtained from the WorldClim Global Climate Data (Hijmans et al. 2005); MAE represents the mean annual potential evapo-transpiration (PET); Tmean represents the monthly mean temperature, TD is calculated as the difference between the

monthly maximum and minimum temperatures; and RA represents the extra-terrestrial radiation on above the atmosphere."

L172 what about bulk density? How was it measured? Reported? Why not use loss-on-ignition?

Response: Bulk density with soil volume measured by coating natural clods in cutting ring then weighing the oven-dried clod in drying oven at 105°C for 24h. Bulk density is calculated by dividing the weight of the oven dried clod by this volume (g·cm$^{-3}$). This way was reported and proved to be credible (Grossman et al., 1968)

How was AGB biomass measured. This is not explained but is an important measure (as indicated by the abstract)

Response: Thanks. We have revised in manuscript as "The plant communty in the subplots was identified, and the above-ground biomass (AGB) was harvested."

L180 is that ration based on mass or volume?

Response: Thanks for your suggestion. We have revised as follow: "The fumigated and nonfumigated samples were extracted using 0.5 M $K_2SO_4$ with a soil:solution mass ratio of 1:4."

L193 what different phases?

Response: Thanks for your comment. Here we mean that there are different phases in the process. Phospholipids were separated from neutral and glycolipids on solid-phase extraction columns by eluting with $CHCl_3$, acetone and methanol, respectively. We have revised as follows: "The resultant fatty acid methyl esters were separated, quantified, and identified using capillary gas chromatography."

L201 why t-test? Maybe an ANOVA should be used to account for the different grassland types? Or was a t-test applied to each type? If the latter, was the p value corrected for multiple tests?

Response: It was ture that this study was conducted on three grassland types, and it was also done along the natural environment gradient (e.g. temperature, precipitation, aridity index) in this grassland transect. Owing to our uneven samping, we conducted the correlation analysis to see the change trend along the environment gradient.

L206 AGB is not defined. L206 provide more details on the source of AI and AGB. What is there resolution? Is there a gradient in the data? Demonstrate that they are gradients. How are they estimated / measured? Provide a description of the data in the results (if they are important variables).

Response: Thanks for your suggestions. We are sorry that we missed this information. We have included more details on the data extraction and data acquisition methods in the revised manuscript. The plant community in

subplots was identified, and the above ground biomass (AGB) was harvested. As to the calculation of AI, we mentioned in the previous reponse.

[Figure]

Figure A1. Geographic locations of the sampling sites in the Inner Mongolian grassland

As Figure A1 shown, our sampling sites were distributed along the aridity index gradient. In Inner Mongolia grasslands, the aridity exhibits a gradient that increases from northeast to southwest (aridity index ranges from 0.16 to 0.54). We have added the Figure A1 to the manuscript.

L214 what is the gradient?

Response: As shown in the Figure A1, the aridity exhibits a gradient that increases from northeast to southwest (aridity index ranges from 0.16 to 0.54). Besides, the study area covered both temperature (mean annual temperature ranges from -2.09 to 7.67) and precipitation (mean annual precipitation ranges from 153.9 to 401.7) gradients.

L216 does distinct mean 'significantly different'
Response: Thanks. Here I mean significant defference betweeen soil depths. We have revised the sentence as "Significantly different water content, soil bulk density, sand percentage and SOC content were found between soil depths (P <0.05, Fig. 1a, 1b, 1c, 1f)."

L216 why is bulk density mentioned here: : : and only here? How was it measured? Did it differ greatly between grassland types?

Response: Bulk density is shown here to show the differences in physicochemical properties between different soil layers. The measurement method was mentioned in the previous reponse.

Why was soil microbial biomass not weighted by bulk density?

Response: In the common way, we performed the usual operation instead of weighting by bulk density. Thanks for your understanding.

L218 the concentrations were larger but was the pool larger? Use the bulk density to evaluate the pool difference.

Response: Here we mean that the concentrations of microbial biomass C, N and P, not the pool. We have revised as "The microbial biomass C, N and P concentrations in the topsoil were significantly higher than that in the subsoil ($P < 0.05$, Table. 2)." Thanks for your understanding.

L225-235 these are very weak significant relationships. This should be acknowledged. Similarly, the relationships in Figure 2 and Figure 3 are not very convincing of a relationship(s).

Response: Thanks for your suggestions. We assume that $R^2$ is good enough to exhibit the change trend. Firstly. The variations in microbial C:N and C:P ratios were partly induced by the measurement method. At the small scale, correlations between fumigation-incubation and fumigation-extraction were variable, which might cause variations in microbial biomass C:N:P stoichiometry (Wardle & Ghani, 1995). Therefore, we assume the variations in microbial biomass C:N:P stoichiometry are inevitable systematic errors.

Second, in a previous study, low $R^2$ also was found along environmental gradients (precipitation, temperature, soil pH, soil content percentage, etc.) at

regional scale (Chen et.al 2016). Finally, several global researches only showed the trend but not even $R^2$ (Xu et al., 2013; Li et al., 2016). This study offered the regional evidence through measurements across a 2100-km$^2$ climatic transect in the Inner Mongolian grasslands.

All in all, we do believe the $R^2$ is good enough to exhibit the trend of microbial C:N along the aridity index gradient. We appreciated that you could accept our explanations.

L225-235 you are regression carbon against a ratio that contains carbon: this is spurious?

Response: Soil organic matter includes the labile (rapid turnover) and stabilized (slow turnover) fractions (Parton et al, 1987). However, the clear and broad consensus is that soil microbes are primarily limited by C availability (Fierer et al. 2003). There is a clear assumption that available C limits biomass and activity (Eilers et al.2012). As soil carbon matter changes, microbial biomass N and microbial biomass P change asymmetrically, which affects the ratios (Mooshammer et al. 2014). Therefore, we assume that the regression of SOC content against microbial C:N:P stoichiometry is a reasonable analysis.

L237 clarify: : : subsoil is reported in L232, L233, and L235.

Response: Thanks for your helpful comments. We have revised this

sentence as: " No or only weak association was found between the microbial C:N, C:P and N:P ratios and the AGB and F:B ratio in the subsoil (Fig. 3)."

L274 drought? Clarify.

Response: Thanks for your helpful comments. As the Table shown, decreasing aridity index means drier weather condition. We have revised the manuscript as follow: "Additionally, microbial C:N ratio decreased with the decreasing aridity index, which serves as a protective mechanism as microbes decrease their nitrogen use efficiency (NUE, the ratio of N invested in growth over total N uptake) and tend to be more N conservative under drier climatic conditions (Mooshammer et al., 2014; Delgado-Baquerizo et al., 2017)." Due to the weak relationship, we have mentioned in the previous reply.

**Table.** Generalized climate classification scheme for *Global*-Aridity values (UNEP 1997)

| Aridity Index Value | Climate Class |
|---|---|
| < 0.03 | Hyper Arid |
| 0.03 – 0.2 | Arid |
| 0.2 – 0.5 | Semi-Arid |
| 0.5 – 0.65 | Dry sub-humid |
| > 0.65 | Humid |

L285 many things change across latitude. Is microbial biomass influenced by latitude or the change in grassland type / climate / etc. Will microbial biomass also change across longitude? What is the range in the aridity gradient in the current study?

Response: In general, latitude pattern was driven by temperature variation. Therefore, we have added results of the temperature analysis to make our conclusions more robust. It is true that microbial biomass exhibits longitudinal pattern in global study (Xu et al., 2013). As shown in Figure A1 and Table 1, aridity index ranges from 0.16 to 0.54 in this study.

L292 this is essentially stating that carbon is related to a ratio that includes carbon. This is not surprising. Is this a spurious (correlation) regression?

Response: Soil organic matter includes the labile (rapid turnover) and stabilized (slow turnover) fractions (Parton et al, 1987). However, the clear and broad consensus is that soil microbes are primarily limited by C availability (Fierer et al., 2003). There is a clear assumption that available C limits biomass and activity (Eilers et al., 2012). As soil carbon matter changes, microbial biomass N and microbial biomass P change asymmetrically, which affects the ratios (Mooshammer et al.,2014). Therefore, we assume that the correlation regression with SOC content is a reasonable analysis.

L309 how were AGB and AI measured? Are they site specific or regional indicators? They only show a weak relationship with little predicative power.

Response: Thanks a lot. The above ground biomass was site-specific while aridity index was a regional indicator. We measured the aboveground biomass

by harvesting the aboveground part of the plants. As to the measurement of AI, more details in the previous reply.

L334 did you quantify spatial heterogeneity? How?

Response: Thanks for your comments. We have removed the speculative statements. The highly variable N:P ratio in microbes may reflect the high variability in site-related P availability (Chen et al. 2013; Li, et al. 2015). Furthermore, the relatively high microbial N:P ratio (suggesting P limitation) are supported by direct evidence showing that low soil P availability strongly limits microbial biomass, activity, and other ecosystem processes (Cleveland et al., 2007). This sentence has been modified as "The high variability of the N:P ratio in soil and soil microbial biomass therefore indicates that the N:P ratio could be an indicator of the ecosystem nutrient status at deeper soil depths (Cleveland et al., 2007; Chen et al. 2013; Li, et al. 2015)."

L337 you cannot answer this question.

Response: Thanks for your comment. We agree that inappropriate statement might result in uncertainty. This sentence has been modified as follows: "How deep should we dig to evaluate the topsoil the microbial stoichiometry in vertical study?"

L341 are the pools distinct?

Response: Thanks a lot. We have removed the speculative statements. We have revised as follows: "The results showed significant differences in the water content and sand percentage, SOC content and F:B ratio between soil depths, suggesting that the resource supplies between topsoil and subsoil were significantly different. "

L350 you tested limited depth, with course increments.

Response: Similar findings were reported in the top 16 cm of soil in a Mediterranean oak forest (0-8 cm and 8-16 cm), where the microbial nutrient ratios (C:N, C:P and N:P) varied between soil depth (Aponte et al., 2010).

L369 what about pools?

Response: Thanks a lot. This study focused on the C:N, C:P and N:P ratios in microbial biomass, not the pools of microbial biomass C, N, P. We don't think that's an important variable.

L375 not shown, this statement is too strongly with respect to drought. There was a weak relationship using a coarse metric.

Response: In addition, microbial C:N ratio decreased with decreasing aridity index, consistent with the perspective that microbes mediate their nitrogen use

efficiency and tend to be more N conservative under drier climatic conditions.

In terms of the weak relationship, we have mentioned in the previous reply.

L383 edaphic? Influence of soil on soil?

Response: Edaphic factor means any characteristic of the environment resulting from the physical, chemical or biotic components of the soil such as the microbial structure, soil texture and soil organic content. In our results, the microbial C:N, C:P and N:P ratios were influenced by SOC content and F:B ratio.

L384 you need to demonstrate the gradient

Response: As the most important gradient, the aridity gradient is demonstrated in Figure A1. Table 1 also showed the ranges of mean annual temperature, mean annual precipitation and above ground biomass in this study.

Reference:
Aponte, C., Marañón, T., García, L. V., Johnson, D., Vile, M., and Wieder, K.: Microbial C, N and P in soils of Mediterranean oak forests: influence of season, canopy cover and soil depth, Biogeochemistry, 101, 77-92, 2010.
Chadwick, O.A., Derry, L.A., Vitousek, P.M., Huebert, B.J., Hedin, L.O., 1999. Changing sources of nutrients during four million years of ecosystem development. Nature 397, 491-497.
Cherwin, K., & Knapp, A. (2012). Unexpected patterns of sensitivity to drought in three semi-arid grasslands. Oecologia, 169(3), 845-852.
Chen, D., J. Cheng, P. Chu, S. Hu, Y. Xie, I. Tuvshintogtokh & Y. Bai (2015) Regional-scale patterns of soil microbes and nematodes across grasslands on the Mongolian plateau: relationships with climate, soil, and plants. Ecography, 38, 622-631.
Chen, Y. L., Chen, L. Y., Peng, Y. F., Ding, J. Z., Li, F., Yang, G. B., Zhang, B. B. (2016). Linking microbial C:N:P stoichiometry to microbial community and abiotic factors along a 3500‐km grassland transect on the Tibetan Plateau. Global Ecology & Biogeography, 25(12), 1416-1427.

Chen, Y., W. Han, L. Tang, Z. Tang & J. Fang (2013) Leaf nitrogen n and phosphorus concentrations of woody plants differ in responses to climate, soil and plant growth form. 36, 178-184.

Cleveland, C. C., and Liptzin, D.: C:N:P Stoichiometry in Soil: Is There a "Redfield Ratio" for the Microbial Biomass?, Biogeochemistry, 85, 235-252, 2007.

Delgado-Baquerizo, M., Powell, J.R., Hamonts, K., Reith, F., Mele, P., Brown, M.V., Dennis, P.G., Ferrari, B.C., Fitzgerald, A., Young, A., 2017. Circular linkages between soil biodiversity, fertilityand plant productivity are limited to topsoil at the continental scale. New Phytologist 215.

Eilers, K. G., Debenport, S., Anderson, S., & Fierer, N. (2012). Digging deeper to find unique microbial communities: The strong effect of depth on the structure of bacterial and archaeal communities in soil. Soil Biology and Biochemistry, 50, 58-65.

Fanin, N., Fromin, N., Buatois, B., Hättenschwiler, S., 2013. An experimental test of the hypothesis of non-homeostatic consumer stoichiometry in a plant litter-microbe system. Ecology Letters 16, 764-772.

Fierer, N., Schimel, J. P., & Holden, P. A. (2003). Variations in microbial community composition through two soil depth profiles. Soil Biology & Biochemistry, 35(1), 167-176.

Joergensen R G.The fumigation-extraction method to estimate soil microbial biomass:Calibration of the Kec value. Soil Biology & Biochemistry,1996,28(1): 25-31

Jenkinson D S, Powlson D S. The effects of biocidal treatments on metabolism in soil—V : A method for measuring soil biomass[J]. Soil Biology & Biochemistry, 1976, 8(3):209-21

Grossman, R. B., Brasher, B. R., Franzmeier, D. P., and Walker, J. L.: Linear Extensibility as Calculated from Natural-Clod Bulk Density Measurements, Soil Science Society of America Journal, 32: 570-573. 32, 570-573,

Hartman, W. H., and Richardson, C. J.: Differential Nutrient Limitation of Soil Microbial Biomass and Metabolic Quotients (q CO2): Is There a Biological Stoichiometry of Soil Microbes?, 8, e57127, 2013.

Hedley, M. J., & Stewart, J. W. B. (1982). Method to measure microbial phosphate in soils. Soil Biology and Biochemistry, 14(4), 377-385.

Hijmans, R.J., Cameron, S.E., Parra, J.L., Jones, P.G. & Jarvis, A. (2004) The WorldClim interpolated global terrestrial climate surfaces, version 1.3

Jaleel, C. A., Manivannan, P., Wahid, A., Farooq, M., Al-Juburi, H. J., Somasundaram, R., International Journal of Agriculture& Biology. (2009). Drought stress in plants: A review on morphological characteristics and pigments composition. 11(1), 100-105.

Kooijman, A. M., Mourik, J. M. V., & Schilder, M. L. M. (2009). The relationship between N mineralization or microbial biomass N with micromorphological properties in beech forest soils with different texture and pH. Biology & Fertility of Soils, 45(5), 449.

Li, P., Yang, Y., Han, W., Fang, J., 2015. Global patterns of soil microbial nitrogen and phosphorus stoichiometry in forest ecosystems. Global Ecology & Biogeography 23, 979-987.

Li, X. Z., and Z. Z. Chen, 2004. Soil microbial biomass C and N along a climatic transect in the Mongolian steppe. Biology & Fertility of Soils 39(5):344-351.

Ross, D. J., and Täte, K. R.: Microbial C and N in litter and soil of a southern beech (Nothofagus) forest: Comparison of measurement procedures, Soil Biology and Biochemistry, 25, 467-475

Manzoni, S., Trofymow, J. A., Jackson, R. B., and Porporato, A.: Stoichiometric controls on carbon, nitrogen, and phosphorus dynamics in decomposing litter, Ecological Monographs, 80, 89-106, 2010.

Maria, M., Wolfgang, W., Sophie, Z. B., and Andreas, R.: Stoichiometric imbalances between terrestrial decomposer communities and their resources: mechanisms and

implications of microbial adaptations to their resources, Frontiers in Microbiology, 5, 22, 2014.

Mooshammer, M., Wanek, W., Hämmerle, I., Fuchslueger, L., Hofhansl, F., Knoltsch, A., Schnecker, J.,Takriti, M., Watzka, M., Wild, B., 2014. Adjustment of microbial nitrogen use efficiency to carbon:nitrogen imbalances regulates soil nitrogen cycling. 5, 3694.

Mouginot, C., Kawamura, R., Matulich, K.L., Berlemont, R., Allison, S.D., Amend, A.S., Martiny, A.C., 2014. Elemental stoichiometry of Fungi and Bacteria strains from grassland leaf litter. Soil Biology & Biochemistry 76, 278-285.

UNEP (1997) World atlas of desertification. United Nations Environment Programme

Parton, W. J., D. S. Schimel, C. V. Cole, and D. S. Ojima. 1987. Analysis of Factors Controlling Soil Organic Matter Levels in Great Plains Grasslands1.Soil Science Society of America Journal J. 51:1173-1179.

Vance E D, Brookes P C, Jenkinson D S. An extraction method for measuring soil microbial biomass C.[J]. Soil Biology & Biochemsitry, 1987, 19(6):703-707

Sterner, R.W., Elser, J.J., 2002. Ecological Stoichiometry: The Biology of Elements From Molecules to The Biosphere.

Wang, C., Wang, X., Liu, D., Wu, H., Lü, X., Fang, Y., Cheng, W., Luo, W., Jiang, P., Shi, J., Yin, H., Zhou, J., Han, X., and Bai, E.: Aridity threshold in controlling ecosystem nitrogen cycling in arid and semi-arid grasslands, Nature Communications, 5, 4799, 10.1038/ncomms5799

Wardle, D. A., & Ghani, A. (1995). Why is the strength of relationships between pairs of methods for estimating soil microbial biomass often so variable? Soil Biology and Biochemistry, 27(6), 821-828.

Wu J S,Joergensen R G, Pommerening B.Measurement of soil microbial biomass C by fumigation extraction An automated procedure .Soil Biology.1990,22(8):1167-1169.

Strickland, M. S., & Rousk, J. (2010). Considering fungal: bacterial dominance in soils – Methods, controls, and ecosystem implications. Soil Biology and Biochemistry, 42(9), 1385-1395.

Tischer, A., Potthast, K., and Hamer, U.: Land-use and soil depth affect resource and microbial stoichiometry in a tropical mountain rainforest region of southern Ecuador, Oecologia, 175, 375-393, 2014.

Trabucco, A., and Zomer, R.J. 2009. Global Aridity Index (Global-Aridity) and Global Potential Evapo-Transpiration (Global-PET) Geospatial Database. CGIAR Consortium for Spatial Information. Published online, available from the CGIAR-CSI Geo Portal at: http://www.csi.cgiar.org

Xu, X., Thornton, P. E., and Post, W. M.: A global analysis of soil microbial biomass carbon, nitrogen and phosphorus in terrestrial ecosystems, Global Ecology & Biogeography, 22, 737– 749, 2013.

Zechmeister-Boltenstern, S., Keiblinger, K. M., Mooshammer, M., Peñuelas, J., Richter, A., Sardans, J., and Wanek, W.: The application of ecological stoichiometry to plant–microbial– soil organic matter transformations, Ecological Monographs, 85, 133-155, 2016.

Zimmerman, A. E., Allison, S. D., & Martiny, A. C. (2014). Phylogenetic constraints on elemental stoichiometry and resource allocation in heterotrophic marine bacteria. 16(5), 1398-1410.

Zomer, R.J., Trabucco, A., Bossio, D.A, van Straaten, O., Verchot, L.V. 2008. Climate Change Mitigation: A Spatial Analysis of Global Land Suitability for Clean Development Mechanism Afforestation and Reforestation. Agric. Ecosystems and Envir. 126: 67-80.